# Multitask Online Mirror Descent

**Nicolò Cesa-Bianchi**                                  *nicolo.cesa-bianchi@unimi.it*
*Department of Computer Science, University of Milan, Italy*

**Pierre Laforgue**                                      *pierre.laforgue@unimi.it*
*Department of Computer Science, University of Milan, Italy*

**Andrea Paudice**                                       *andrea.paudice@unimi.it*
*Department of Computer Science, University of Milan, Italy*
*Istituto Italiano di Tecnologia, Genova, Italy*

**Massimiliano Pontil**                                  *massimiliano.pontil@iit.it*
*Istituto Italiano di Tecnologia, Genova, Italy*
*University College London, United Kingdom*

**Reviewed on OpenReview:** *https://openreview.net/forum?id=zwRX9kkKzj*

## Abstract

We introduce and analyze MT-OMD, a multitask generalization of Online Mirror Descent (OMD) which operates by sharing updates between tasks. We prove that the regret of MT-OMD is of order $\sqrt{1 + \sigma^2(N-1)}\sqrt{T}$, where $\sigma^2$ is the task variance according to the geometry induced by the regularizer, $N$ is the number of tasks, and $T$ is the time horizon. Whenever tasks are similar, that is $\sigma^2 \leq 1$, our method improves upon the $\sqrt{NT}$ bound obtained by running independent OMDs on each task. We further provide a matching lower bound, and show that our multitask extensions of Online Gradient Descent and Exponentiated Gradient, two major instances of OMD, enjoy closed-form updates, making them easy to use in practice. Finally, we present experiments which support our theoretical findings.

## 1 Introduction

In multitask learning (Caruana, 1997), one faces a set of tasks to solve, and tries to leverage their similarities to learn faster. Task similarity is often formalized in terms of Euclidean distances among the best performing models for each task, see Evgeniou & Pontil (2004) for an example. However, in online convex optimization, and Online Mirror Descent (OMD) in particular, it is well known that using different geometries to measure distances in the model space can bring substantial advantages — see, e.g., Hazan (2016); Orabona (2019). For instance, when the model space is the probability simplex in $\mathbb{R}^d$, running OMD with the KL divergence (corresponding to an entropic regularizer) allows one to learn at a rate depending only logarithmically on $d$. It is thus natural to investigate to what extent measuring task similarities using geometries that are possibly non-Euclidean could improve the analysis of online multitask learning. From an application perspective, typical online multitask scenarios include federated learning applications for mobile users (e.g., personalized recommendation or health monitoring) or for smart homes (e.g., energy consumption prediction), mobile sensor networks for environmental monitoring, or even networked weather forecasting. These scenarios fit well with online learning, as new data is being generated all the time, and require different losses and decision sets, motivating the design of a general framework.

In this work, we introduce MT-OMD, a multitask generalization of OMD which applies to any strongly convex regularizer. We present a regret analysis establishing that MT-OMD outperforms OMD (run independently on each task) whenever tasks are similar according to the geometry induced by the regularizer. Our work

builds on the multitask extension of the Perceptron algorithm developed in Cavallanti et al. (2010), where prior knowledge about task similarities is expressed through a symmetric positive definite interaction matrix $A$. Typically, $A = I + L$, where $L$ is the Laplacian of a task relatedness graph with adjacency matrix $W$. The authors then show that the number of mistakes depends on $\sum_i \|u_i\|_2^2 + \sum_{i,j} W_{ij}\|u_i - u_j\|_2^2$, where each $u_i$ denotes the best model for task $i$. This expression can be seen as a measure of task dispersion with respect to matrix $W$ and norm $\|\cdot\|_2$. The Euclidean norm appears because the Perceptron is an instance of OMD for the hinge loss with the Euclidean regularizer, so that distances in the model space are measured through the corresponding Bregman divergence, which is the Euclidean squared norm.

For an arbitrary strongly convex regularizer $\psi$, the regret of OMD is controlled by a Bregman divergence and a term inversely proportional to the curvature of the regularizer. The key challenge we face is how to extend the OMD regularizer to the multitask setting so that the dispersion term captures task similarities. A natural strategy would be to choose a regularizer whose Bregman divergence features $\sum_{i,j} W_{ij} B_\psi(u_i, u_j)$. Although this mimics the Euclidean dispersion term of the Perceptron, the associated regularizer has a small curvature, compromising the divergence-curvature balance which, as we said, controls the regret. Observing that the Perceptron's dispersion term can be rewritten $\|\boldsymbol{A}^{1/2}\boldsymbol{u}\|_2^2$, where $\boldsymbol{A}$ is a block version (across tasks) of $A$ and $\boldsymbol{u}$ is the concatenation of the reference vectors $u_i$, our solution consists in using the regularizer $\boldsymbol{\psi}(\boldsymbol{A}^{1/2}\cdot)$, where $\boldsymbol{\psi}$ is the compound version of any base regularizer $\psi$ defined on the model space. While exhibiting the right curvature, this regularizer has still the drawback that $\boldsymbol{A}^{1/2}\boldsymbol{u}$ might be outside the domain of $\boldsymbol{\psi}$. To get around this difficulty, we introduce a notion of variance aligned with the geometry induced by $\boldsymbol{\psi}$, such that the corresponding Bregman divergence $B_{\boldsymbol{\psi}}(\boldsymbol{A}^{1/2}\boldsymbol{u}, \boldsymbol{A}^{1/2}\boldsymbol{v})$ is always defined for sets of tasks with small variance. We then show that the Bregman divergence can be upper bounded in terms of the task variance $\sigma^2$, and by tuning appropriately the matrix $A$ we obtain a regret bound for MT-OMD that scales as $\sqrt{1 + \sigma^2(N-1)}$. In contrast, the regret of independent OMD scales as $\sqrt{N}$, highlighting the advantage brought by MT-OMD when tasks have a small variance. We stress that this improvement is independent of the chosen regularizer, thereby offering a substantial acceleration in a wide range of scenarios. To keep the exposition simple, we first work with a fixed and known $\sigma^2$. We then show an extension of MT-OMD that does not require any prior knowledge on the task similarity. The rest of the paper is organized as follows. In Section 2, we introduce the multitask online learning problem and describe MT-OMD, our multitask extension to solve it. In Section 3, we derive a regret analysis for MT-OMD, which highlights its advantage when tasks are similar. Section 4 is devoted to algorithmic implementations, and Section 5 to experiments.

**Related work.** Starting from the seminal work by Caruana (1997), multitask learning has been intensively studied for more than two decades, see Zhang & Yang (2021) for a recent survey. Similarly to Cavallanti et al. (2010), our work is inspired by the Laplacian multitask framework of Evgeniou et al. (2005). This framework has been extended to kernel-based learning (Sheldon, 2008), kernel-based unsupervised learning (Gu et al., 2011), contextual bandits (Cesa-Bianchi et al., 2013), spectral clustering (Yang et al., 2014), stratified model learning (Tuck et al., 2021), and, more recently, federated learning (Dinh et al., 2021). See also Herbster & Lever (2009) for different applications of Laplacians in online learning. A multitask version of OMD has been previously proposed by Kakade et al. (2012). Their approach, unlike ours, is cast in terms of matrix learning, and uses group norms and Schatten $p$-norm regularizers. Their bounds scale with the diameter of the model space according to these norms (as opposed to scaling with the task variance, as in our analysis). Moreover, their learning bias is limited to the choice of the matrix norm regularizer and does not explicitly include a notion of task similarity matrix. Abernethy et al. (2007); Dekel et al. (2007) investigate different multitask extensions of online learning, see also Alquier et al. (2017); Finn et al. (2019); Balcan et al. (2019); Denevi et al. (2019) for related extensions to meta-learning. Some online multitask applications are studied in Pillonetto et al. (2008); Li et al. (2014; 2019), but without providing any regret analyses. Saha et al. (2011); Zhang et al. (2018) extend the results of Cavallanti et al. (2010) to dynamically updated interaction matrices. However, no regret bounds are provided. Murugesan et al. (2016) look at distributed online classification and prove regret bounds, but they are not applicable in our asynchronous model. Other approaches for learning task similarities include Zhang & Yeung (2010); Pentina & Lampert (2017); Shui et al. (2019). We finally note the recent work by Boursier et al. (2022), which establishes multitask learning guarantees with trace norm regularization when the number of samples per task is small, and that by Laforgue et al. (2022), which learns jointly the tasks and their structure, but only with the Euclidean regularizer and under the assumption that the task activations are stochastic.

Although our asynchronous multitask setting is identical to that of Cavallanti et al. (2010), we emphasize that our work extends theirs much beyond the fact that we consider arbitrary convex losses instead of just the hinge loss. Algorithmically, MT-OMD generalizes the Multitask Perceptron in much the same way OMD generalizes the standard Perceptron. From a technical point of view, Theorem 1 in Cavallanti et al. (2010) is a direct consequence of the Kernel Perceptron Theorem, and is therefore limited to Euclidean geometries. Instead, our work provides a complete analysis of all regularizers of the form $\psi(A^{1/2}\cdot)$. Although Cavallanti et al. (2010) also contains a non-Euclidean $p$-norm extension of the Multitask Perceptron, we point out that their extension is based on a regularizer of the form $\|Au\|_p^2$. This is different from MT-OMD for $p$-norms, which instead uses $\sum_i \|(A^{1/2}u)^{(i)}\|_p^2$. As a consequence, their bound is worse than ours (see Appendix C for technical details), does not feature any variance term, and does not specialize to the Euclidean case when $p = 2$. Note that our analysis on the simplex is also completely novel as far as we know.

## 2 Multitask Online Learning

We now describe the multitask online learning problem, and introduce our approach to solve it. We use a cooperative and asynchronous multiagent formalism: the online algorithm is run in a distributed fashion by communicating agents, that however make predictions at different time steps.

**Problem formulation and reminders on OMD.** We consider an online convex optimization setting with a set of $N \in \mathbb{N}$ agents, each learning a possibly different task on a common convex decision set $V \subset \mathbb{R}^d$. At each time step $t = 1, 2, \ldots$ some agent $i_t \leq N$ makes a prediction $x_t \in V$ for its task, incurs loss $\ell_t(x_t)$, and observes a subgradient of $\ell_t$ at $x_t$, where $\ell_t$ is a convex loss function. We say that $i_t$ is the active agent at time $t$. Both the sequence $i_1, i_2, \ldots$ of active agents and the sequence $\ell_1, \ell_2, \ldots$ of convex losses are chosen adversarially and hidden from the agents. Note that the algorithm we propose is deterministic, such that $\ell_t$ might be indifferently chosen before $x_t$ is predicted (oblivious adversary) or after. Our goal is to minimize the *multitask regret*, which is defined as the sum of the individual regrets

$$R_T = \sum_{i=1}^{N} \left( \sum_{t:\, i_t=i} \ell_t(x_t) - \inf_{u \in V} \sum_{t:\, i_t=i} \ell_t(u) \right) = \sum_{t=1}^{T} \ell_t(x_t) - \sum_{i=1}^{N} \inf_{u \in V} \sum_{t:\, i_t=i} \ell_t(u). \tag{1}$$

A natural idea to minimize Equation (1) is to run $N$ independent OMDs, one for each agent. Recall that OMD refers to a family of algorithms, typically used to minimize a regret of the form $\sum_t \ell_t(x_t) - \inf_{u \in V} \sum_t \ell_t(u)$, for any sequence of proper convex loss functions $\ell_t$. An instance of OMD is parameterized by a $\lambda$-strongly convex regularizer $\psi \colon \mathbb{R}^d \to \mathbb{R}$, and has the update rule

$$x_{t+1} = \arg\min_{x \in V} \ \langle \eta_t g_t, x \rangle + B_\psi(x, x_t), \tag{2}$$

where $g_t \in \mathbb{R}^d$ is a subgradient of $\ell_t$ at point $x_t$, $B_\psi(x, y) = \psi(x) - \psi(y) - \langle \nabla \psi(y), x - y \rangle$ denotes the Bregman divergence associated to $\psi$, and $\eta_t > 0$ is a tunable learning rate. Standard results allow to bound the regret achieved by the sequence of iterates produced by OMD. For a fixed $\eta$ and any initial point $x_1 \in V$, we have (Orabona, 2019, Theorem 6.8) that for all $u \in V$

$$\sum_{t=1}^{T} \ell_t(x_t) - \ell_t(u) \leq \frac{B_\psi(u, x_1)}{\eta} + \frac{\eta}{2\lambda} \sum_{t=1}^{T} \|g_t\|_\star^2, \tag{3}$$

with $\|\cdot\|_\star$ the dual norm of the norm with respect to which $\psi$ is strongly convex (see Definition 4 in the Appendix). The choice of the regularizer $\psi$ shapes the above bound through the quantities $B_\psi(u, x_1)$ and $\|g_t\|_\star$. When $\psi = \frac{1}{2}\|\cdot\|_2^2$, we have $B_\psi(x, y) = \frac{1}{2}\|x - y\|_2^2$, $\|\cdot\|_\star = \|\cdot\|_2$, $\lambda = 1$, and the algorithm is called Online Gradient Descent (OGD). However, depending on the problem, a different choice of the regularizer might better captures the underlying geometry. A well-known example is Exponentiated Gradient (EG), an instance of OMD in which $V$ is the probability simplex in $\mathbb{R}^d$, such that $V = \Delta := \{x \in \mathbb{R}_+^d \colon \sum_j x_j = 1\}$. EG uses the negative entropy regularizer $x \mapsto \sum_j x_j \ln(x_j)$, and assuming that $\|g_t\|_\infty \leq L_g$, one achieves bounds of order $\mathcal{O}(L_g\sqrt{T \ln d})$, while OGD yields bounds of order $\mathcal{O}(L_g\sqrt{dT})$. We emphasize that our cooperative

extension adapts to several types of regularizers, and can therefore exploit these improvements with respect to the dependence on $d$, see Proposition 8. Let $C$ be a generic constant such that $C\sqrt{T}$ bounds the regret incurred by the chosen OMD (e.g., $C = L_g\sqrt{\ln d}$, or $C = L_g\sqrt{d}$ above). Then, by Jensen's inequality the multitask regret of $N$ independent OMDs satisfies

$$R_T \le \sum_{i=1}^{N} C\sqrt{T_i} \le C\sqrt{NT}\,, \tag{4}$$

where $T_i = \sum_{t=1}^{T} \mathbb{I}\{i_t = i\}$ denotes the number of times agent $i$ was active. Our goal is to show that introducing communication between the agents may significantly improve on Equation (4) with respect to the dependence on $N$.

**A multitask extension.** We now describe our multitask OMD approach. To gain some insights on it, we first focus on OGD. For $i \le N$ and $t \le T$, let $x_{i,t} \in \mathbb{R}^d$ denote the prediction maintained by agent $i$ at time step $t$. By completing the square in Equation (2) for $\psi = \psi_{\text{Euc}} := \frac{1}{2}\|\cdot\|_2^2$, the independent OGDs updates can be rewritten for all $i \le N$ and $t$ such that $i_t = i$:

$$x_{i,t+1} = \Pi_{V,\|\cdot\|_2}\big(x_{i,t} - \eta_t g_t\big)\,, \tag{5}$$

where $\Pi_{V,\|\cdot\|}$ denotes the projection operator onto the convex set $V$ according to the norm $\|\cdot\|$, that is $\Pi_{V,\|\cdot\|}(x) = \arg\min_{y \in V}\|x - y\|$. Our analysis relies on *compound representations*, that we explain next. We use bold notation to refer to compound vectors, such that for $u_1, \dots, u_N \in \mathbb{R}^d$, the compound vector is $\boldsymbol{u} = [u_1, \dots, u_N] \in \mathbb{R}^{Nd}$. For $i \le N$, we use $\boldsymbol{u}^{(i)}$ to refer to the $i^{\text{th}}$ block of $\boldsymbol{u}$, such that $\boldsymbol{u}^{(i)} = u_i$ in the above example. So $\boldsymbol{x}_t$ is the compound vector of the $(x_{i,t})_{i=1}^{N}$, such that $x_t = \boldsymbol{x}_t^{(i_t)}$, and the multitask regret rewrites as $R_T(\boldsymbol{u}) = \sum_{t=1}^{T} \ell_t\big(\boldsymbol{x}_t^{(i_t)}\big) - \ell_t\big(\boldsymbol{u}^{(i_t)}\big)$. For any set $V \subset \mathbb{R}^d$, let $\boldsymbol{V} = V^{\otimes N} \subset \mathbb{R}^{Nd}$ denote the compound set such that $\boldsymbol{u} \in \boldsymbol{V}$ is equivalent to $\boldsymbol{u}^{(i)} \in V$ for all $i \le N$. Equipped with this notation, the independent OGD updates Equation (5) rewrite as

$$\boldsymbol{x}_{t+1} = \Pi_{\boldsymbol{V},\|\cdot\|_2}\big(\boldsymbol{x}_t - \eta_t \boldsymbol{g}_t\big)\,, \tag{6}$$

with $\boldsymbol{g}_t \in \mathbb{R}^{Nd}$ such that $\boldsymbol{g}_t^{(i)} = g_t$ for $i = i_t$, and $0_{\mathbb{R}^d}$ otherwise. In other words, only the active agent has a non-zero gradient and therefore makes an update. Our goal is to incorporate communication into this independent update. To that end, we consider the general idea of *sharing updates* by considering (sub) gradients of the form $\boldsymbol{A}^{-1}\boldsymbol{g}_t$, where $\boldsymbol{A}^{-1} \in \mathbb{R}^{Nd \times Nd}$ is a shortcut notation for $A^{-1} \otimes I_d$ and $A \in \mathbb{R}^{N \times N}$ is any symmetric positive definite interaction matrix. Note that $A$ is a parameter of the algorithm playing the role of a learning bias. While our central result (Theorem 1) holds for any choice of $A$, our more specialized bounds (see Propositions 5 to 8) apply to a parameterized family of matrices $A$. A simple computation shows that $(\boldsymbol{A}^{-1}\boldsymbol{g}_t)^{(i)} = A_{ii_t}^{-1} g_t$. Thus, every agent $i$ makes an update proportional to $A_{ii_t}^{-1}$ at each time step $t$. In other words, the active agent (the only one to suffer a loss) shares its update with the other agents. Results in Section 3 are proved by designing a matrix $A^{-1}$ (or equivalently $A$) such that $A_{ii_t}^{-1}$ captures the similarity between tasks $i$ and $i_t$. Intuitively, the active agent $i_t$ should share its update (gradient) with another agent $i$ to the extent their respective tasks are similar. Overall, denoting by $\|u\|_M = \sqrt{u^\top M u}$ the Mahalanobis norm of $u$, the MT-OGD update writes

$$\boldsymbol{x}_{t+1} = \Pi_{\boldsymbol{V},\|\cdot\|_{\boldsymbol{A}}}\big(\boldsymbol{x}_t - \eta_t \boldsymbol{A}^{-1}\boldsymbol{g}_t\big)\,. \tag{7}$$

In comparison to Equation (6), the need for changing the norm in the projection, although unclear at first sight, can be explained in multiple ways. First, it is key to the analysis, as we see in the proof of Theorem 1. Second, it can be interpreted as another way of exchanging information between agents, see Remark 1. Finally, note that update Equation (7) can be decomposed as

$$\begin{aligned}
\widetilde{\boldsymbol{x}}_{t+1} &= \operatorname*{arg\,min}_{\boldsymbol{x} \in \mathbb{R}^{Nd}} \ \langle \eta_t \boldsymbol{g}_t, \boldsymbol{x}\rangle + \frac{1}{2}\|\boldsymbol{x} - \boldsymbol{x}_t\|_{\boldsymbol{A}}^2\,, \\
\boldsymbol{x}_{t+1} &= \operatorname*{arg\,min}_{\boldsymbol{x} \in \boldsymbol{V}} \ \frac{1}{2}\|\boldsymbol{x} - \widetilde{\boldsymbol{x}}_{t+1}\|_{\boldsymbol{A}}^2\,,
\end{aligned} \tag{8}$$

showing that it is natural to keep the same norm in both updates. Most importantly, what Equation (8) tells us, is that the MT-OGD update rule is actually an OMD update—see e.g., (Orabona, 2019, Section 6.4)—with regularizer $\boldsymbol{x} \mapsto \frac{1}{2}\|\boldsymbol{x}\|_{\boldsymbol{A}}^2 = \boldsymbol{\psi}_{\mathrm{Euc}}(\boldsymbol{A}^{1/2}\boldsymbol{x})$. This provides a natural path for extending our multitask approach to any regularizer. Given a base regularizer $\psi \colon \mathbb{R}^d \to \mathbb{R}$, the *compound regularizer* $\boldsymbol{\psi}$ is given by $\boldsymbol{\psi} \colon \boldsymbol{x} \in \mathbb{R}^{Nd} \mapsto \sum_{i=1}^N \psi(\boldsymbol{x}^{(i)})$. When there exists a function $\phi \colon \mathbb{R} \to \mathbb{R}$ such that $\psi(x) = \sum_j \phi(x_j)$, the compound regularizer is the natural extension of $\psi$ to $\mathbb{R}^{Nd}$. Note, however, that the relationship can be more complex, e.g., when $\psi(x) = \frac{1}{2}\|x\|_p^2$. Using regularizer $\boldsymbol{\psi}(\boldsymbol{A}^{1/2} \cdot)$, whose associate divergence is $B_{\boldsymbol{\psi}}(\boldsymbol{A}^{1/2}\boldsymbol{x}, \boldsymbol{A}^{1/2}\boldsymbol{x}')$, the MT-OMD update thus reads

$$\boldsymbol{x}_{t+1} = \underset{\boldsymbol{x} \in \boldsymbol{V}}{\arg\min} \ \langle \eta_t \boldsymbol{g}_t, \boldsymbol{x}\rangle + B_{\boldsymbol{\psi}}(\boldsymbol{A}^{1/2}\boldsymbol{x}, \boldsymbol{A}^{1/2}\boldsymbol{x}_t). \tag{9}$$

Clearly, if $\psi = \psi_{\mathrm{Euc}}$, we recover the MT-OGD update. Observe also that whenever $A = I_N$, MT-OMD is equivalent to $N$ independent OMDs. We conclude this exposition with a remark shedding light on the way MT-OMD introduces communication between agents.

**Remark 1** (Communication mechanism in MT-OMD). *Denoting $\boldsymbol{y}_t = \boldsymbol{A}^{1/2}\boldsymbol{x}_t$, Equation (9) rewrites*

$$\boldsymbol{x}_{t+1} = \boldsymbol{A}^{-1/2} \underset{\boldsymbol{y} \in \boldsymbol{A}^{1/2}(\boldsymbol{V})}{\arg\min} \ \langle \eta_t \boldsymbol{A}^{-1/2}\boldsymbol{g}_t, \boldsymbol{y}\rangle + B_{\boldsymbol{\psi}}(\boldsymbol{y}, \boldsymbol{y}_t). \tag{10}$$

*The two occurrences of $\boldsymbol{A}^{-1/2}$ reveal that agents communicate in two distinct ways: one through the shared update (the innermost occurrence of $\boldsymbol{A}^{-1/2}$), and one through computing the final prediction $\boldsymbol{x}_{t+1}$ as a linear combination of the solution to the optimization problem. Multiplying Equation (10) by $\boldsymbol{A}^{1/2}$, MT-OMD can also be seen as standard OMD on the transformed iterate $\boldsymbol{y}_t$.*

## 3 Regret Analysis

We now provide a regret analysis for MT-OMD. We start with a general theorem presenting two bounds, for constant and time-varying learning rates. These results are then instantiated to different types of regularizer and variance in Propositions 2 to 8. The main difficulty is to characterize the strong convexity of $\boldsymbol{\psi}(\boldsymbol{A}^{1/2} \cdot)$, see Lemmas 11 and 12 in the Appendix. Throughout the section, $V \subset \mathbb{R}^d$ is a convex set of comparators, and $(\ell_t)_{t=1}^T$ is a sequence of proper convex loss functions chosen by the adversary. Note that all technical proofs can be found in Appendix A.

**Theorem 1.** *Let $\psi \colon \mathbb{R}^d \to \overline{\mathbb{R}}$ be $\lambda$-strongly convex with respect to norm $\|\cdot\|$ on $V$, let $A \in \mathbb{R}^{N \times N}$ be symmetric positive definite, and set $\boldsymbol{x}_1 \in \boldsymbol{V}$. Then, MT-OMD with $\eta_t := \eta$ produces a sequence of iterates $(\boldsymbol{x}_t)_{t=1}^T$ such that for all $\boldsymbol{u} \in \boldsymbol{V}$, $R_T(\boldsymbol{u})$ is bounded by*

$$\frac{B_{\boldsymbol{\psi}}(\boldsymbol{A}^{1/2}\boldsymbol{u}, \boldsymbol{A}^{1/2}\boldsymbol{x}_1)}{\eta} \ + \ \max_{i \leq N} A_{ii}^{-1} \ \frac{\eta}{2\lambda} \sum_{t=1}^T \|g_t\|_\star^2. \tag{11}$$

*Moreover, for any sequence of nonincreasing learning rates $(\eta_t)_{t=1}^T$, MT-OMD produces a sequence of iterates $(\boldsymbol{x}_t)_{t=1}^T$ such that for all $\boldsymbol{u} \in \boldsymbol{V}$, $R_T(\boldsymbol{u})$ is bounded by*

$$\max_{t \leq T} \frac{B_{\boldsymbol{\psi}}(\boldsymbol{A}^{1/2}\boldsymbol{u}, \boldsymbol{A}^{1/2}\boldsymbol{x}_t)}{\eta_T} \ + \ \max_{i \leq N} A_{ii}^{-1} \ \frac{1}{2\lambda} \sum_{t=1}^T \eta_t \|g_t\|_\star^2. \tag{12}$$

### 3.1 Multitask Online Gradient Descent

For $\psi = \frac{1}{2}\|\cdot\|_2^2$, $A = I_N$ (independent updates), unit-norm reference vectors $(\boldsymbol{u}^{(i)})_{i=1}^N$, $L_g$-Lipschitz losses, and $\boldsymbol{x}_1 = \boldsymbol{0}$, bound Equation (11) becomes: $ND^2/2\eta + \eta T L_g^2/2$. Choosing $\eta = D\sqrt{N}/L_g\sqrt{T}$, we recover the $DL_g\sqrt{NT}$ bound of Equation (4). Our goal is to design interaction matrices $A$ that make Equation (11) smaller. In the absence of additional assumptions on the set of comparators, it is however impossible to get a systematic improvement: the bound is a sum of two terms, and introducing interactions typically reduces one term but increases the other. To get around this difficulty, we introduce a simple condition on the task similarity, that allows us to control the increase of $B_{\boldsymbol{\psi}}(\boldsymbol{A}^{1/2}\boldsymbol{u}, \boldsymbol{A}^{1/2}\boldsymbol{x}_1)$ for a carefully designed class of interaction matrices.

**Definition 1.** *Let* $\|\cdot\| \colon \mathbb{R}^d \to \mathbb{R}$ *be any norm, and* $\bar{\boldsymbol{u}} = (1/N)\sum_{i=1}^{N} \boldsymbol{u}^{(i)}$, *for any* $\boldsymbol{u} \in \mathbb{R}^{Nd}$. *We define the variance of* $\boldsymbol{u}$ *w.r.t.* $\|\cdot\|$ *as*

$$\mathrm{Var}_{\|\cdot\|}(\boldsymbol{u}) = \frac{1}{N-1}\sum_{i=1}^{N}\left\|\boldsymbol{u}^{(i)} - \bar{\boldsymbol{u}}\right\|^2.$$

*Let* $D = \sup_{u \in V}\|u\|$, *and* $\sigma > 0$. *The comparators with variance smaller than* $\sigma^2 D^2$ *are denoted by*

$$\boldsymbol{V}_{\|\cdot\|,\sigma} = \left\{\boldsymbol{u} \in \boldsymbol{V} : \mathrm{Var}_{\|\cdot\|}(\boldsymbol{u}) \le \sigma^2 D^2\right\}. \tag{13}$$

For sets of comparators of the form Equation (13), we show that MT-OGD achieves significant improvements over its independent counterpart. The rationale behind this gain is fairly natural: the tasks associated with comparators in Equation (13) are similar due to the variance constraint, so that communication indeed helps. Note that condition Equation (13) does not enforce any restriction on the norms of the individual $\boldsymbol{u}^{(i)}$, and is much more complex than a simple rescaling of the feasible set by $\sigma^2$. For instance, one could imagine task vectors highly concentrated around some vector $u_0$, whose norm is $D$: the individual norms are close to $D$, but the task variance is small. This is precisely the construction used in the separation result (Proposition 4). As MT-OMD leverages the additional information of the task variance (unavailable in the independent case), it is expected that an improvement *should be* possible. The problems of how to use this extra information and what improvement can be achieved through it are addressed in the rest of this section. To that end, we first assume $\sigma^2$ to be known. This assumption can be seen as a learning bias, analog to the knowledge of the diameter $D$ in standard OGD bounds. In Section 3.4, we then detail a Hedge-based extension of MT-OGD that does not require the knowledge of $\sigma^2$ and only suffers an additional regret of order $\sqrt{T \log N}$.

The class of interaction matrices we consider is defined as follows. Let $L = I_N - \mathbb{1}\mathbb{1}^\top/N$. We consider matrices of the form $A(b) = I_N + bL$, where $b \ge 0$ quantifies the magnitude of the communication. For more intuition about this choice, see Section 3.4. We can now state a first result highlighting the advantage brought by MT-OGD.

**Proposition 2.** *Let* $\psi = \frac{1}{2}\|\cdot\|_2^2$, $D = \sup_{x \in V}\|x\|_2$, *and* $\sigma \le 1$. *Assume that* $\|\partial \ell_t(x)\|_2 \le L_g$ *for all* $t \le T$ *and any* $x \in V$. *Set* $b = N$, $\boldsymbol{x}_1 = \boldsymbol{0}$, *and* $\eta = D\sqrt{N(N+1)(1+(N-1)\sigma^2)}/L_g\sqrt{2T}$. *Then, MT-OGD produces a sequence of iterates* $(\boldsymbol{x}_t)_{t=1}^T$ *such that for all* $\boldsymbol{u} \in \boldsymbol{V}_{\|\cdot\|_2,\sigma}$

$$R_T(\boldsymbol{u}) \le DL_g\sqrt{1 + \sigma^2(N-1)}\sqrt{2T}. \tag{14}$$

*Proof sketch.* With $\psi = \frac{1}{2}\|\cdot\|_2^2$, and $\boldsymbol{x}_1 = \boldsymbol{0}$, we have $2B_\psi\big(\boldsymbol{A}(b)^{1/2}\boldsymbol{u}, \boldsymbol{A}(b)^{1/2}\boldsymbol{0}\big) = \|\boldsymbol{u}\|_2^2 + b(N-1)\mathrm{Var}_{\|\cdot\|_2}(\boldsymbol{u})$, which is smaller than $ND^2(1 + b\frac{N-1}{N}\sigma^2)$. Then, it is easy to check that $\big[A(b)^{-1}\big]_{ii} = \frac{b+N}{(1+b)N}$ for all $i \le N$. Substituting these values into Equation (11), we obtain

$$R_T(\boldsymbol{u}) \le \frac{ND^2(1 + b\frac{N-1}{N}\sigma^2)}{2\eta} + \frac{\eta T L_g^2}{2}\frac{b+N}{(1+b)N}.$$

Finally, set $\eta = \frac{ND}{L_g}\sqrt{\frac{\left(1 + b\frac{N-1}{N}\sigma^2\right)(1+b)}{(b+N)T}}$ and $b = N$. $\qquad\square$

Thus, MT-OGD enjoys a $\sqrt{1 + \sigma^2(N-1)}$ dependence, which is smaller than $\sqrt{N}$ when tasks have a variance smaller than 1. When $\sigma = 0$ (all tasks are equal), MT-OGD scales as if there were only one task. When $\sigma \ge 1$, the analysis suggests to choose $b = 0$, i.e., $A = I_N$, and one recovers the performance of independent OGDs. Note that the additional $\sqrt{2}$ factor in Equation (14) can be removed for limit cases through a better optimization in $b$: the bound obtained in the proof actually reads $DL\sqrt{F(\sigma)T}$, with $F(0) = 1$ and $F(1) = N$. However, the function $F$ lacks of interpretability outside of the limit cases (for details see Appendix A.2) motivating our choice to present the looser but more interpretable bound Equation (14). For a large $N$, we have $\sqrt{1 + \sigma^2(N-1)} \approx \sigma\sqrt{N}$. The improvement brought by MT-OGD is thus roughly proportional to the square root of the task variance. From now on, we refer to this gain as the *multitask acceleration*. This improvement achieved by MT-OGD is actually optimal up to constants, as revealed by the following lower bound, which is only 1/4 of Equation (14).

**Proposition 3.** *Under the conditions of Proposition 2, the regret of any algorithm satisfies*

$$\sup_{\boldsymbol{u} \in \boldsymbol{V}_{\|\cdot\|_2,\sigma}} R_T(\boldsymbol{u}) \geq \frac{1}{4} \left( D L_g \sqrt{1 + \sigma^2(N-1)} \sqrt{2T} \right).$$

Another way to gain intuition about Equation (14) is to compare it to the lower bound for OGD considering independent tasks (IT-OGD). The following separation result shows that MT-OGD may strictly improve over IT-OGD.

**Proposition 4.** *Let $d \geq 9$, $N = 2d$, and $\sigma \leq 1$ to be tuned later. Then, there exists $\boldsymbol{u} \in \boldsymbol{V}_{\|\cdot\|_2,\sigma}$ such that*

$$R_T^{\mathrm{IT-OGD}}(\boldsymbol{u}) \geq \frac{\sqrt{(1 - 2\sigma^2)N}}{4} \sqrt{2T}.$$

*Proposition 2 then yields that for any $\sigma^2 < \frac{N-16}{18N-16}$*

$$R_T^{\mathrm{IT-OGD}}(\boldsymbol{u}) > R_T^{\mathrm{MT-OGD}}(\boldsymbol{u}).$$

### 3.2 Extension to any Norm Regularizers

A natural question is: *can the multitask acceleration be achieved with other regularizers?* Indeed, the proof of Proposition 2 crucially relies on the fact that the Bregman divergence can be exactly expressed in terms of $\|\boldsymbol{u}\|_2^2$ and $\mathrm{Var}_{\|\cdot\|_2}(\boldsymbol{u})$. In the following proposition, we show that such an improvement is also possible for all regularizers of the form $\frac{1}{2}\|\cdot\|^2$, for arbitrary norms $\|\cdot\|$, up to an additional multiplicative constant. A crucial application is the use of the $p$-norm on the probability simplex, which is known to exhibit a logarithmic dependence in $d$ for a well-chosen $p$.

**Proposition 5.** *Let $\|\cdot\|\colon \mathbb{R}^d \to \mathbb{R}$ be any norm, $\psi = \frac{1}{2}\|\cdot\|^2$, $D = \sup_{x \in V}\|x\|$, and $\sigma \leq 1$. Assume that $\|\partial \ell_t(x)\|_\star \leq L_g$ for all $t \leq T$, $x \in V$. Set $b = N$, $\boldsymbol{x}_1 = \boldsymbol{0}$ and $\eta = D\sqrt{N(N+1)(1+(N-1)\sigma^2)}/L_g\sqrt{2T}$. Then, MT-OMD produces a sequence of iterates $(\boldsymbol{x}_t)_{t=1}^T$ such that for all $\boldsymbol{u} \in \boldsymbol{V}_{\|\cdot\|,\sigma}$*

$$R_T(\boldsymbol{u}) \leq D L_g \sqrt{1 + \sigma^2(N-1)} \sqrt{8T}.$$

*In particular, for $d \geq 3$ and $V = \Delta$, choosing $\|\cdot\| = \|\cdot\|_p$, for $p = 2\ln d/(2\ln d - 1)$, and assuming that $\|\partial \ell_t(x)\|_\infty \leq L_g$, it holds for all $\boldsymbol{u} \in \boldsymbol{\Delta}_{\|\cdot\|_p,\sigma}$*

$$R_T(\boldsymbol{u}) \leq L_g \sqrt{1 + \sigma^2(N-1)} \sqrt{16e\, T \ln d}.$$

*In comparison, under the same assumptions, bound Equation (14) would write as: $L_g\sqrt{1 + \sigma^2(N-1)}\sqrt{2Td}$.*

**Projecting onto $\boldsymbol{V}_\sigma$.** Propositions 2 and 5 reveal that whenever tasks are similar (i.e., whenever $\boldsymbol{u} \in \boldsymbol{V}_\sigma$), then using the regularizer $\boldsymbol{\psi}(\boldsymbol{A}^{1/2}\,\cdot\,)$ with $A \neq I_N$ accelerates the convergence. However, this is not the only way to leverage the small variance condition. For instance, one may also use this information to directly project onto $\boldsymbol{V}_\sigma \subset \boldsymbol{V}$, by considering the update

$$\boldsymbol{x}_{t+1} = \underset{\boldsymbol{x} \in \boldsymbol{V}_\sigma}{\arg\min} \ \langle \eta_t \boldsymbol{g}_t, \boldsymbol{x} \rangle + B_{\boldsymbol{\psi}}\left(\boldsymbol{A}^{1/2}\boldsymbol{x}, \boldsymbol{A}^{1/2}\boldsymbol{x}_t\right). \tag{15}$$

Although not necessary in general (Propositions 2 and 5 show that communicating the gradients is sufficient to get an improvement), this refinement presents several advantages. First, it might be simpler to compute in practice, see Section 4. Second, it allows for adaptive learning rates, that preserve the guarantees while being independent from the horizon $T$ (Proposition 6). Finally, it allows to derive $L^\star$ bounds with the multitask acceleration for smooth loss functions (Proposition 7). Results are stated for arbitrary norms, but bounds sharper by a factor 2 can be obtained for $\|\cdot\|_2$.

**Proposition 6.** *Let $\|\cdot\|\colon \mathbb{R}^d \to \mathbb{R}$ be any norm, $\psi = \frac{1}{2}\|\cdot\|^2$, $D = \sup_{x \in V}\|x\|$, and $\sigma \le 1$. Set $b = N$, and $\eta_t = D\sqrt{N(N+1)(1+(N-1)\sigma^2)}(\sum_{i=1}^t \|g_i\|_\star^2)^{-1/2}$. Then, Equation (15) produces a sequence of iterates $(\boldsymbol{x}_t)_{t=1}^T$ such that for all $\boldsymbol{u} \in \boldsymbol{V}_{\|\cdot\|,\sigma}$*

$$R_T(\boldsymbol{u}) \le 8D\sqrt{1+\sigma^2(N-1)}\left(\sum_{t=1}^T \|g_t\|_\star^2\right)^{1/2}.$$

**Proposition 7.** *Let $\|\cdot\|\colon \mathbb{R}^d \to \mathbb{R}$ be any norm, $\psi = \frac{1}{2}\|\cdot\|^2$, $D = \sup_{x \in V}\|x\|$, and $\sigma \le 1$. Assume that the $\ell_t$ are $M$-smooth, i.e., $\|\nabla \ell_t(x) - \nabla \ell_t(y)\|_\star \le M\|x-y\|$ for all $t \le T$, and any $x, y \in V$. Set $b$ and $\eta_t$ as in Proposition 6. Then, update Equation (15) produces a sequence of iterates $(\boldsymbol{x}_t)_{t=1}^T$ such that for all $\boldsymbol{u} \in \boldsymbol{V}_{\|\cdot\|,\sigma}$*

$$R_T(\boldsymbol{u}) \le 16D\sqrt{1+\sigma^2(N-1)}\left(2MD\sqrt{1+\sigma^2(N-1)} + \sqrt{M\sum_{t=1}^T \ell_t\big(\boldsymbol{u}^{(i_t)}\big)}\right).$$

**Remark 2** (Strongly convex and exp-concave losses). *Another popular assumption to derive improved regret bounds is to consider strongly convex or exp-concave losses. Recall that a function $f$ is said to be $\alpha$-exp-concave if $\exp(-\alpha f)$ is concave (for instance, the logistic loss of a linear predictor with norm bounded by $U$ and unit-norm inputs is $\exp(-2U)/2$-exp-concave). In this context, standard single task algorithms achieve improved regret bounds of order $\ln T$, see e.g., (Orabona, 2019, Corollary 7.24 and Section 7.10). We highlight that a simple adaptation of the single task analysis is not enough to exhibit a multitask acceleration in these cases. Indeed, the cornerstone of our analysis is to leverage the compound representation, in which the interactions are more easily analyzed. On the other hand, the strong convexity or exp-concavity of the losses provide a sharper control on the instantaneous regret that depends on the norm of the active predictor/comparator only, but cannot be extended to the compound framework. Another way to look at the problem is to recall that FTRL deals with strongly convex losses by choosing $\psi \equiv 0$. This means $\widetilde{\psi} = \psi(\boldsymbol{A}^{1/2}\cdot) \equiv 0$, such that MT-FTRL is equivalent to independent FTRL and no multitask improvement can be achieved.*

### 3.3  Regularizers on the Simplex

As seen in Propositions 2 to 7, MT-OMD induces a multitask acceleration in a wide range of settings, involving different regularizers (Euclidean norm, $p$-norms) and various kind of loss functions (Lipschitz continuous, smooth continuous gradients). This systematic gain suggests that multitask acceleration essentially derives from our approach, and is completely orthogonal to the improvements achievable by choosing the regularizer appropriately. Bounds combining both benefits are actually derived in the second claim of Proposition 5. However, all regularizers studied so far share a crucial feature: they are defined on the entire space $\mathbb{R}^d$. As a consequence, the divergence $B_{\boldsymbol{\psi}}(\boldsymbol{A}^{1/2}\boldsymbol{u}, \boldsymbol{A}^{1/2}\boldsymbol{x})$ is always well defined, which might not be true in general, for instance when the comparator set studied is the probability simplex $\Delta$. A workaround consists in assigning the value $+\infty$ to the Bregman divergence whenever either of the arguments is outside of the compound simplex $\boldsymbol{\Delta} = \Delta^{\otimes N}$. The choice of the interaction matrix $A$ then becomes critical to prevent the bound from exploding, and calls for a new definition of the variance. Indeed, note that for $i \le N$ we have $(\boldsymbol{A}(b)^{1/2}\boldsymbol{u})^{(i)} = \sqrt{1+b}\,\boldsymbol{u}^{(i)} + (1-\sqrt{1+b})\bar{\boldsymbol{u}}$. If all $\boldsymbol{u}^{(i)}$ are equal (say to $u_0 \in \Delta$), then all $(\boldsymbol{A}(b)^{1/2}\boldsymbol{u})^{(i)}$ are also equal to $u_0$ and $\boldsymbol{A}(b)^{1/2}\boldsymbol{u} \in \boldsymbol{\Delta}$. However, if they are different, by definition of $\bar{\boldsymbol{u}}$, for all $j \le d$, there exists $i \le N$ such that $\boldsymbol{u}_j^{(i)} \le \bar{\boldsymbol{u}}_j$. Then, for $b$ large enough, $\sqrt{1+b}\,\boldsymbol{u}_j^{(i)} + (1-\sqrt{1+b})\bar{\boldsymbol{u}}_j$ becomes negative, and $(\boldsymbol{A}(b)^{1/2}\boldsymbol{u})^{(i)}$ is out of the simplex. Luckily, the maximum acceptable value for $b$ can be easily deduced from the following variance definition.

**Definition 2.** *Let $\boldsymbol{u} \in \mathbb{R}^d$. For all $j \le d$, let*

$$\boldsymbol{u}_j^{\max} = \max_{i \le N} \boldsymbol{u}_j^{(i)}, \quad and \quad \boldsymbol{u}_j^{\min} = \min_{i \le N} \boldsymbol{u}_j^{(i)}.$$

*Then, with the convention $0/0 = 0$ we define*

$$\mathrm{Var}_\Delta(\boldsymbol{u}) = \max_{j \le d}\left(\frac{\boldsymbol{u}_j^{\max} - \boldsymbol{u}_j^{\min}}{\boldsymbol{u}_j^{\max}}\right)^2,$$

*and for any $\sigma \leq 1$*

$$\boldsymbol{\Delta}_\sigma = \left\{ \boldsymbol{u} \in \boldsymbol{\Delta} \colon \mathrm{Var}_\Delta(\boldsymbol{u}) \leq \sigma^2 \right\}.$$

Equipped with this new variance definition, we can now analyze regularizers defined on the simplex.

**Proposition 8.** *Let $\psi : \Delta \to \mathbb{R}$ be $\lambda$-strongly convex w.r.t. norm $\|\cdot\|$, and such that there exist $x^* \in \Delta$ and $C < +\infty$ such that for all $x \in \Delta, B_\psi(x, x^*) \leq C$. Let $\sigma \leq 1$, and assume that $\|\partial \ell_t(x)\|_\star \leq L_g$ for all $t \leq T$ and $x \in \Delta$. Set $b = (1 - \sigma^2)/\sigma^2$, $\boldsymbol{x}_1 = [x^*, \dots, x^*]$, and $\eta = N\sqrt{2\lambda(1 + b)C}/L_g\sqrt{(b + N)T}$. Then, MT-OMD produces a sequence of iterates $(\boldsymbol{x}_t)_{t=1}^T$ such that for all $\boldsymbol{u} \in \boldsymbol{\Delta}_\sigma$*

$$R_T(\boldsymbol{u}) \leq L_g\sqrt{1 + \sigma^2(N - 1)}\sqrt{2CT/\lambda}.$$

*For the negative entropy we have $x^* = \mathbb{1}/d$ and $C = \ln d$. With subgradients satisfying $\|\partial \ell_t(x)\|_\infty \leq L_g$ we obtain*

$$R_T(\boldsymbol{u}) \leq L_g\sqrt{1 + \sigma^2(N - 1)}\sqrt{2T \ln d}.$$

Proposition 8 shows that the multitask acceleration is not an artifact of the Euclidean geometry, but rather a general feature of MT-OMD, as long as the variance definition is aligned with the geometry of the problem.

## 3.4 Adaptivity to the Task Variance

Most of the results we presented so far require the knowledge of the task variance $\sigma^2$. We now present an Hedge-based extension of MT-OMD, denoted Hedge-MT-OMD, that does not require any prior information on $\sigma^2$. First, note that for $\sigma^2 \geq 1$, MT-OMD becomes equivalent to independent OMDs. A simple approach consists then in using Hedge—see, e.g., (Orabona, 2019, Section 6.8)—over a set of experts, each running an instance of MT-OMD with a different value of $\sigma^2$ chosen on a uniform grid of the interval $[0, 1]$.[1] We can show that Hedge-MT-OGD only suffers an additional regret of order $\sqrt{T \log N}$ against MT-OGD run with the exact knowledge of $\mathrm{Var}_{\|\cdot\|_2}(\boldsymbol{u})$.

**Theorem 9.** *Let $D = \sup_{x \in V} \|x\|_2$, and assume that $\|\partial \ell_t(x)\|_2 \leq L_g$ for all $t \leq T$, $x \in V$. Then, for all $\boldsymbol{u} \in \boldsymbol{V}$ the regret of Hedge-MT-OGD is bounded by*

$$DL_g\left(2 + \sqrt{\log N} + \sqrt{\min\left\{\mathrm{Var}_{\|\cdot\|_2}(\boldsymbol{u}), 1\right\} \cdot N}\right)\sqrt{2T}.$$

**Variance definition and choice of $A$.** Note that we have $(N - 1)\mathrm{Var}_{\|\cdot\|_2}(\boldsymbol{u}) = \frac{1}{N}\sum_{i,j}\|\boldsymbol{u}^{(i)} - \boldsymbol{u}^{(j)}\|_2^2 = \boldsymbol{u}^\top \boldsymbol{L}\boldsymbol{u}$, where $L = I_N - \mathbb{1}\mathbb{1}^\top/N$ is the Laplacian of the weighted clique graph over $\{1, \dots, N\}$, with edges of $1/N$. A natural extension then consists in considering variances of the form

$$\mathrm{Var}_{\|\cdot\|_2}^W(\boldsymbol{u}) = \sum_{i,j=1}^N W_{ij}\|\boldsymbol{u}^{(i)} - \boldsymbol{u}^{(j)}\|_2^2 = \boldsymbol{u}^\top \boldsymbol{L}^W \boldsymbol{u}$$

for any adjacency matrix $W$ and its Laplacian $L^W$. For instance, if we expect tasks to be concentrated in clusters, it is natural to consider $W_{ij} = 1$ if $\boldsymbol{u}^{(i)}$ and $\boldsymbol{u}^{(j)}$ (are thought to) belong to the same cluster, and 0 otherwise. This local version is interesting, as it allows to satisfy the variance condition with a smaller $\sigma$, which improves the MT-OMD regret bound. Note that the proof of Theorem 1 can be readily adapted to this definition by considering the class of interaction matrices $\{A(b) = I_N + bL^W\}$. The bound however features $\max_{i \leq N}[A(b)^{-1}]_{ii}$, which depends on $W$ in a nontrivial way and requires a case by case analysis, preventing from stating a general result for an arbitrary $W$. Considering even more general matrices $A$, i.e., that do not write as $I_N + bL$, suffers from the same problem (one then also needs to compute $B_\psi(\boldsymbol{A}^{1/2}\boldsymbol{u}, \boldsymbol{A}^{1/2}\boldsymbol{v})$ on a case by case basis), and does not enjoy anymore the variance interpretation seen above. Furthermore, note that Proposition 2 is obtained by minimizing Equation (11) with respect to $A$. For matrices of the form $A(b)$, this tradeoff only depends on $b$, and is thus much easier to solve than for general matrices. Finally, we stress that local variances can be similarly considered on the simplex. Instead of involving the global $\boldsymbol{u}_j^{\max}$, the variance formula then features for each task/node a local maximum (respectively minimum) over its neighbours.

---

[1] Note that Hedge-MT-OGD computes the loss subgradient at arbitrary points (corresponding to the expert's predictions).

### 3.5 Additional Remarks

We conclude this section with two additional results. The first one draws an interesting connection between our multitask framework and the notion of dynamic regret, while the second emphasizes on the importance of the asynchronous nature of the activations.

**Remark 3** (Connection to dynamic regret). *Note that our online multitask setting can be viewed as a special case of dynamic regret, where each comparator $u_t$ in the sequence of comparators $u_1, u_2, \ldots$ belongs to an unknown set $\{u'_1, \ldots, u'_N\}$ of known cardinality $N$. Moreover, at the beginning of each time step $t$, the learner is told the index $i_t \in \{1, \ldots, N\}$ of the comparator $u_t$ against which the regret is measured at time $t$. As a consequence, any algorithm for dynamic regret minimization can be used in our setting. In the Euclidean case, the optimal dynamic regret bound is of order $\sqrt{D^2 + DV_T}\sqrt{T}$, where $D$ is the Euclidean diameter of the decision space and $V_T = \sum_{t=1}^{T} \|u_{t+1} - u_t\|_2$. In order to facilitate the comparison to our bound $\sqrt{D^2 + D^2(N-1)\sigma^2}\sqrt{T}$, let assume that the sequence of adversarial activations is such that $i_t \neq i_{t-1}$, and that all pairs of distinct elements in $\{u'_1, \ldots, u'_N\}$ appear with the same frequency as consecutive comparators $u_t, u_{t+1}$. Let $P_T = \sum_{t=1}^{T} \|u_{t+1} - u_t\|_2^2$. We have*

$$P_T = \sum_{t=1}^{T} \|u_{t+1} - u_t\|_2^2 = \frac{T}{N(N-1)} \sum_{i \neq j} \|u'_i - u'_j\|_2^2 = 2T \frac{1}{N(N-1)} \sum_{i \neq j} \frac{\|u'_i - u'_j\|_2^2}{2} = 2T\sigma^2 D^2,$$

*such that our upper bound is at most*

$$\sqrt{D^2 + D^2(N-1)\sigma^2}\sqrt{T} = \sqrt{D^2 + (N-1)\frac{P_T}{2T}}\sqrt{T} \leq \sqrt{D^2 + \frac{N-1}{2T}DV_T}\sqrt{T},$$

*which is better as soon as $T \geq \frac{N-1}{2}$.*

We finally derive a regret bound in the case where several agents are active at each time step.

**Proposition 10.** *Consider the setting of Proposition 2, but assume now that at each time step $t$ a subset of agents $\mathcal{A}_t \subset \{1, \ldots, N\}$, of cardinality $|\mathcal{A}_t| = p$, is chosen by the adversary and asked to make predictions. Recall that in this case the regret of an independent approach is of order $DL_g\sqrt{pNT}$. Then, MT-OMD run with $b = \sqrt{p}N$ and $\eta = ND\sqrt{1 + \sqrt{p}\sigma^2(N-1)}/L_g\sqrt{p(1+p)T}$ produces a sequence of iterates $(\boldsymbol{x}_t)_{t=1}^{T}$ such that for all $\boldsymbol{u} \in \boldsymbol{V}_{\|\cdot\|_2,\sigma}$*

$$R_T(\boldsymbol{u}) \leq DL_g\sqrt{pT}\sqrt{1 + \sigma^2(N-1)}\sqrt{p^{1/2} + p^{3/2}}.$$

*Note that for $p = 1$, we recover exactly Proposition 2.*

Proposition 10 highlights that having asynchronous activations is critical: the more active agents at a single time step, i.e., the bigger $p$, the bigger $\sqrt{1 + \sigma^2(N-1)}\sqrt{p^{1/2} + p^{3/2}}$, i.e., the smaller the multitask acceleration. In the extreme case where $p = N$, our multitask framework reduces to a standard online convex optimization problem, with diameter $\sqrt{N}D$ and gradients with Euclidean norms bounded by $\sqrt{N}L_g$. Standard lower bounds are then of order $NDL_g\sqrt{T}$, confirming that no multitask acceleration is possible.

## 4 Algorithms

We now show that MT-OGD and MT-EG enjoy closed-form updates, making them easy to implement. Note that the MT-OGD derivation is valid for any matrix $A$ positive definite, while MT-EG requires $A^{-1/2}$ to be stochastic. This is verified by matrices of the form $A = I_N + L^W$ (Lemma 13).

**MT-OGD.** Let $V = \{u \in \mathbb{R}^d : \|u\|_2 \leq D\}$, and $\sigma \leq 1$. Recall that $\boldsymbol{V} = V^{\otimes N} = \{\boldsymbol{u} \in \mathbb{R}^{Nd} : \|\boldsymbol{u}\|_{2,\infty} \leq D\}$, and $\boldsymbol{V}_{\|\cdot\|_2,\sigma} = \{\boldsymbol{u} \in \boldsymbol{V} : \text{Var}_{\|\cdot\|_2}(\boldsymbol{u}) \leq \sigma^2\}$. Solving the first equation in Equation (8), we obtain that the iterate $\boldsymbol{x}_{t+1}$ produced by MT-OGD is the solution to

$$\min_{\boldsymbol{x} \in \mathbb{R}^{Nd}} \left\{ \left\| \boldsymbol{x}_t - \eta_t \boldsymbol{A}^{-1} \boldsymbol{g}_t - \boldsymbol{x} \right\|_{\boldsymbol{A}}^2 \; : \; \|\boldsymbol{x}\|_{2,\infty} \leq D \right\}.$$

However, computing this update is made difficult by the discrepancy between the norms used in the objective and the constraint. A simple work around consists in considering the minimization over the Mahalanobis ball $\boldsymbol{V_A} = \{\boldsymbol{u} \in \mathbb{R}^{Nd} \colon \|\boldsymbol{u}\|_{\boldsymbol{A}}^2 \leq (1 + b\sigma^2)ND^2\}$ instead. It is easy to check that $\boldsymbol{V}_{\|\cdot\|_2, \sigma} \subset \boldsymbol{V_A}$, so that every result derived in Section 3 for MT-OGD remains valid (only the fact that comparators and iterates are in $\boldsymbol{V_A}$ is actually used). With the substitution $\boldsymbol{y}_t = \boldsymbol{A}^{1/2}\boldsymbol{x}_t$ the MT-OGD update then rewrites (see Appendix B.1 for technical details)

$$\boldsymbol{y}_{t+1} = \text{Proj}\Big(\boldsymbol{y}_t - \eta_t \boldsymbol{A}^{-1/2}\boldsymbol{g}_t, \sqrt{(1 + b\sigma^2)N}D\Big), \tag{16}$$

where $\text{Proj}(x, \tau) = \min\big\{1, \frac{\tau}{\|x\|_2}\big\}x$. Note that Equation (16) can be easily turned back into an update on $\boldsymbol{x}_t$ by making the inverse substitution. In practice however, $\boldsymbol{x}_t$ is only computed to make the predictions. The pseudo-code of the algorithm is given in Algorithm 1.

---

**Algorithm 1** MT-OGD

---

**input:** A positive definite matrix $A \in \mathbb{R}^{N \times N}$ (interaction matrix), $\sigma^2 > 0$ (task variance), $D > 0$ (comparators set diameter), $\eta_t > 0$ (learning-rate schedule)
**initialization:** $\boldsymbol{A} = A \otimes I_d \in \mathbb{R}^{Nd \times Nd}$, $\boldsymbol{x}_1 = \boldsymbol{0}$, $\boldsymbol{y}_1 = \boldsymbol{A}^{-1/2}\boldsymbol{x}_1$
 1: Compute $\boldsymbol{A}^{1/2}$ and $\boldsymbol{A}^{-1/2}$
 2: **for** time steps $t = 1, 2, \ldots$ **do**
 3:     Compute compound prediction $\boldsymbol{x}_t = \boldsymbol{A}^{1/2}\boldsymbol{y}_t$
 4:     Activate agent $i_t$ and predict $x_t^{(i_t)}$
 5:     Get $g_t \in \partial\ell(x_t)$ and compute $\boldsymbol{g}_t$
 6:     Compute $\boldsymbol{y}_{t+1} = \min\left\{1, \frac{\sqrt{(1+b\sigma^2)N}D}{\|\boldsymbol{y}_t - \eta_t \boldsymbol{A}^{-1/2}\boldsymbol{g}_t\|_2}\right\}(\boldsymbol{y}_t - \eta_t \boldsymbol{A}^{-1/2}\boldsymbol{g}_t)$
 7: **end for**

---

**MT-EG.** Using Equation (8) with $\boldsymbol{y}_t = \boldsymbol{A}^{1/2}\boldsymbol{x}_t$, MT-EG reads

$$\tilde{\boldsymbol{y}}_{t+1} = \underset{\boldsymbol{y} \in \mathbb{R}^{Nd}}{\arg\min} \ \langle \eta \boldsymbol{A}^{-1/2}\boldsymbol{g}_t, \boldsymbol{y}\rangle + B_{\boldsymbol{\psi}}(\boldsymbol{y}, \boldsymbol{y}_t),$$

$$\boldsymbol{y}_{t+1} = \underset{\boldsymbol{y} \in \boldsymbol{A}^{1/2}(\boldsymbol{\Delta})}{\arg\min} \ B_{\boldsymbol{\psi}}(\boldsymbol{y}, \tilde{\boldsymbol{y}}_{t+1}), \tag{17}$$

where $\boldsymbol{\psi}$ is the compound negative entropy regularizer such that $\boldsymbol{\psi}(\boldsymbol{x}) = \sum_{i=1}^N \sum_{j=1}^d \boldsymbol{x}_j^{(i)} \ln\big(\boldsymbol{x}_j^{(i)}\big)$. One can show (see Appendix B.2 for details) that the update can be rewritten for all $i \leq N$ and $j \leq d$

$$\tilde{\boldsymbol{y}}_{t+1,j}^{(i)} = \boldsymbol{y}_{t,j}^{(i)} \ \exp\left(-\eta A_{ii_t}^{-1/2} g_{t,j} - 1\right),$$

$$\boldsymbol{y}_{t+1,j}^{(i)} = \frac{\tilde{\boldsymbol{y}}_{t+1,j}^{(i)}}{\sum_{k=1}^d \tilde{\boldsymbol{y}}_{t+1,k}^{(i)}}.$$

Combining both equations, we finally obtain

$$\boldsymbol{y}_{t+1,j}^{(i)} = \frac{\boldsymbol{y}_{t,j}^{(i)} \ e^{-\eta A_{ii_t}^{-1/2} g_{t,j}}}{\sum_{k=1}^d \boldsymbol{y}_{t,k}^{(i)} \ e^{-\eta A_{ii_t}^{-1/2} g_{t,k}}}. \tag{18}$$

Update Equation (18) enjoys a natural interpretation. Each block $\boldsymbol{y}^{(i)}$ is operating an individual standard EG update, but with gradient $A_{ii_t}^{-1/2} g_t$. When $A = I_N$, only the active block is updated. Otherwise, the update of block $i$ is proportional to $A_{ii_t}^{-1/2}$, that quantifies the similarity between tasks $i$ and $i_t$. The pseudo-code of the algorithm is given in Algorithm 2. Although this work only focuses on OMD for clarity, note that considering Follow-the-Regularized-Leader—see, e.g., (Orabona, 2019, Section 7)—with $\widetilde{\boldsymbol{\psi}} = \boldsymbol{\psi}(\boldsymbol{A}^{1/2}\cdot)$ would yield similar bounds. This would allow, for instance, the use of time-varying learning rates with entropic regularization.

---

**Algorithm 2** MT-EG

---

**input:** A positive definite matrix $A \in \mathbb{R}^{N \times N}$ (interaction matrix), $\eta > 0$ (learning-rate)
**initialization:** $\boldsymbol{A} = A \otimes I_d \in \mathbb{R}^{Nd \times Nd}$, $\boldsymbol{x}_1 = \mathbb{1}/d \in \mathbb{R}^{Nd}$, $\boldsymbol{y}_1 = \boldsymbol{A}^{-1/2}\boldsymbol{x}_1$

1: Compute $\boldsymbol{A}^{1/2}$ and $\boldsymbol{A}^{-1/2}$
2: **for** time steps $t = 1, 2, \ldots$ **do**
3:      Compute compound prediction $\boldsymbol{x}_t = \boldsymbol{A}^{1/2}\boldsymbol{y}_t$
4:      Activate agent $i_t$ and predict $x_t^{(i_t)}$
5:      Get $g_t \in \partial\ell(x_t)$ and compute $\boldsymbol{g}_t$
6:      Compute $\boldsymbol{y}_{t+1,j}^{(i)} = \dfrac{\boldsymbol{y}_{t,j}^{(i)} \ e^{-\eta A_{i i_t}^{-1/2} g_{t,j}}}{\sum_{k=1}^{d} \boldsymbol{y}_{t,k}^{(i)} \ e^{-\eta A_{i i_t}^{-1/2} g_{t,k}}}, \forall i \in [N], \forall j \in [d]$
7: **end for**

---

## 5 Experiments

In this section, we empirically compare the performance of Hedge-MT-OGD/EG against two natural alternatives: an independent-task approach (IT-OGD/EG) where the agents do not communicate, and a single-task approach (ST-OGD/EG) where a single model is learned and shared by all agents. Note that both IT and ST approaches are special cases of MT-OMD, obtained respectively with the choices $b = 0$ (i.e., $\sigma^2 \geq 1$), or $b = +\infty$ (i.e., $\sigma^2 = 0$). In Appendix D we report an additional experiment where we empirically validate the dependence of the performance of MT-OGD on the task variance.

**Online Gradient Descent.** For this experiment, we use the *Lenk* dataset (Lenk et al., 1996; Argyriou et al., 2007). It consists of 2880 computer ratings in the range $\{1, 2, \ldots, 10\}$, made by 180 individuals (the tasks) on the basis of 14 binary features. Each computer is rated on a discrete scale from 0 to 10, expressing the likelihood of an individual buying that computer. We run Hedge-MT-OGD using the clique interaction matrix $A = (1 + N)I_N - \mathbb{1}\mathbb{1}^\top$ and the square loss. For all algorithms, the value of $\eta$ is set according to the optimal theoretical value, see Proposition 2. In Hedge-MT-OGD, the variance $\sigma^2$ is learned in a set of 5 experts uniformly spaced over $[0, 1]$. For simplicity, we use $D = 1$ and compute the resulting Lipschitz constant accordingly. Results are reported in Figure 1(a).

**Exponentiated Gradient.** For our second experiment, we consider *EMNIST*, a classification dataset consisting of 62 classes (images of digits, small and capital letters). To speed up computation, we reduced the number of features from 784 down to 10 through a standard dimensionality reduction method. We created 61 binary classification tasks by considering the 0 digit class against each other class. To each task, we assigned 10 examples (5 positive, 5 negative) randomly chosen from the set of examples for that task. We considered the linear logistic regression and ran Hedge-MT-EG with the parameterized clique interaction matrix $A(b) = (1 + b)I_N - b\mathbb{1}\mathbb{1}^\top/N$. The value of $b$ is set according to the theoretical value (that depends on $\sigma^2$, see Proposition 8), while $\sigma^2$ is learned in a set of 5 experts uniformly spaced over $[0, 1]$. For all algorithms, the value of $\eta$ is set according to the optimal theoretical values. Results are reported in Figure 1(b).

## 6 Conclusion

We introduced and analyzed MT-OMD, a multitask extension of OMD whose regret is shown to improve as the task variance, expressed in terms of the geometry induced by the regularizer, decreases. We provided a unifying analysis and a single algorithm that explains when is multitask acceleration possible based on the current geometry, and how to achieve it. Natural and interesting directions for future research include: (1) analyzing the multitask acceleration in combination with other properties, such as strongly convex losses, and (2) designing and analyzing an extension of MT-OMD that is adaptive to the best interaction matrix.

## References

Jacob Abernethy, Peter Bartlett, and Alexander Rakhlin. Multitask learning with expert advice. In *Proceedinds of the 20th International Conference on Computational Learning Theory*, pp. 484–498, 2007.

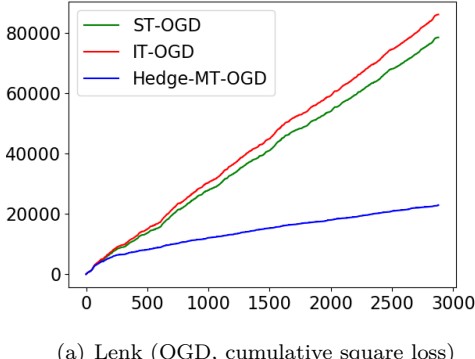

(a) Lenk (OGD, cumulative square loss)

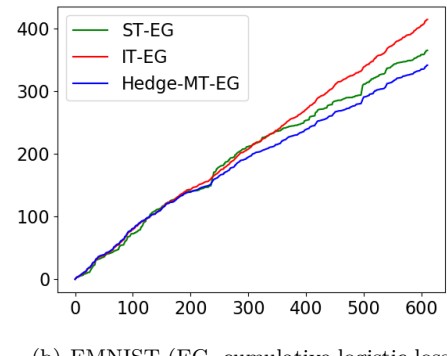

(b) EMNIST (EG, cumulative logistic loss)

Figure 1: Comparison between multitask (MT), independent-task (IT), and single-task (ST) OGD and EG on the *Lenk* and *EMNIST* datasets. We plot the cumulative losses against time. *Lenk* is known to work well in multi-task settings, and indeed Hedge-MT-OGD performs significantly better than both baselines. On the other hand, *EMNIST* has a variance significantly higher than *Lenk*. However, even in this unfavorable scenario, Hedge-MT-EG is still outperforming the baselines, though by a small margin.

Gotz Alefeld and Norbert Schneider. On square roots of m-matrices. *Linear Algebra and its Applications*, 42: 119–132, 1982.

Pierre Alquier, The Tien Mai, and Massimiliano Pontil. Regret bounds for lifelong learning. In *Proceedinds of the 20th International Conference Artificial Intelligence and Statistics*, pp. 261–269, 2017.

Andreas Argyriou, Charles A Micchelli, Massimiliano Pontil, and Yiming Ying. A spectral regularization framework for multi-task structure learning. In *Proceedings of the 20th Annual Conference on Neural Information Processing Systems*, pp. 25–32, 2007.

Maria-Florina Balcan, Mikhail Khodak, and Ameet Talwalkar. Provable guarantees for gradient-based meta-learning. In *Proceedinds of the 36th International Conference on Machine Learning*, pp. 424–433, 2019.

Heinz H. Bauschke and Patrick L. Combettes. *Convex analysis and monotone operator theory in Hilbert spaces.* Springer, 2011.

Etienne Boursier, Mikhail Konobeev, and Nicolas Flammarion. Trace norm regularization for multi-task learning with scarce data. *arXiv preprint arXiv:2202.06742*, 2022.

Stephen Boyd, Stephen P Boyd, and Lieven Vandenberghe. *Convex Optimization.* Cambridge University Press, 2004.

Rich Caruana. Multitask learning. *Machine Learning*, 28(1):41–75, 1997.

Giovanni Cavallanti, Nicolò Cesa-Bianchi, and Claudio Gentile. Linear algorithms for online multitask classification. *Journal of Machine Learning Research*, 11:2901–2934, 2010.

Nicolò Cesa-Bianchi, Claudio Gentile, and Giovanni Zappella. A gang of bandits. In *Proceedings of the 26th Annual Conference on Neural Information Processing Systems*, pp. 737–745, 2013.

Nicolò Cesa-Bianchi and Gábor Lugosi. *Prediction, learning, and games.* Cambridge University Press, 2006.

Ofer Dekel, Philip M Long, and Yoram Singer. Online learning of multiple tasks with a shared loss. *Journal of Machine Learning Research*, 8(10):2233–2264, 2007.

Giulia Denevi, Dimitris Stamos, Carlo Ciliberto, and Massimiliano Pontil. Online-within-online meta-learning. In *Proceedings of the 32th Annual Conference on Advances in Neural Information Processing Systems 32*, pp. 13089–13099, 2019.

Canh T Dinh, Tung T Vu, Nguyen H Tran, Minh N Dao, and Hongyu Zhang. Fedu: A unified framework for federated multi-task learning with Laplacian regularization. *arXiv preprint arXiv:2102.07148*, 2021.

Theodoros Evgeniou and Massimiliano Pontil. Regularized multi–task learning. In *Proceedings of the 10th ACM SIGKDD International Conference on Knowledge Discovery and Data Mining*, pp. 109–117, 2004.

Theodoros Evgeniou, Charles A Micchelli, and Massimiliano Pontil. Learning multiple tasks with kernel methods. *Journal of Machine Learning Research*, 6(4):615–637, 2005.

Chelsea Finn, Aravind Rajeswaran, Sham Kakade, and Sergey Levine. Online meta-learning. In *Proceedings of the 36th International Conference on Machine Learning*, pp. 1920–1930, 2019.

Claudio Gentile. The robustness of the p-norm algorithms. *Machine Learning*, 53(3):265–299, 2003.

Adam J Grove, Nick Littlestone, and Dale Schuurmans. General convergence results for linear discriminant updates. In *Proceedings 10th Annual Conference on Computational Learning Theory*, pp. 171–183, 1997.

Quanquan Gu, Zhenhui Li, and Jiawei Han. Learning a kernel for multi-task clustering. In *Proceedings of the 25th AAAI Conference on Artificial Intelligence*, pp. 368–373, 2011.

Elad Hazan. Introduction to online convex optimization. *Foundations and Trends in Optimization*, 2(3-4):157–325, 2016.

Mark Herbster and Guy Lever. Predicting the labelling of a graph via minimum p-seminorm interpolation. In *Proceedings of the 22nd Conference on Learning Theory*, 2009.

Sham M Kakade, Shai Shalev-Shwartz, and Ambuj Tewari. Regularization techniques for learning with matrices. *Journal of Machine Learning Research*, 13(1):1865–1890, 2012.

Pierre Laforgue, Andrea Della Vecchia, Nicolò Cesa-Bianchi, and Lorenzo Rosasco. Adatask: Adaptive multitask online learning. *arXiv preprint arXiv:2205.15802*, 2022.

Peter J Lenk, Wayne S DeSarbo, Paul E Green, and Martin R Young. Hierarchical bayes conjoint analysis: Recovery of partworth heterogeneity from reduced experimental designs. *Marketing Science*, 15(2):173–191, 1996.

Guangxia Li, Steven CH Hoi, Kuiyu Chang, Wenting Liu, and Ramesh Jain. Collaborative online multitask learning. *IEEE Transactions on Knowledge and Data Engineering*, 26(8):1866–1876, 2014.

Rui Li, Fenglong Ma, Wenjun Jiang, and Jing Gao. Online federated multitask learning. In *Proceedings of the 7th IEEE International Conference on Big Data*, pp. 215–220, 2019.

Keerthiram Murugesan, Hanxiao Liu, Jaime Carbonell, and Yiming Yang. Adaptive smoothed online multi-task learning. In *Proceedings of the 29th Annual Conference on Advances in Neural Information Processing Systems*, pp. 4296–4304, 2016.

Francesco Orabona. A modern introduction to online learning. *arXiv preprint arXiv:1912.13213*, 2019.

Anastasia Pentina and Christoph H Lampert. Multi-task learning with labeled and unlabeled tasks. In *Proceedings of the 34th International Conference on Machine Learning*, pp. 2807–2816, 2017.

Gianluigi Pillonetto, Francesco Dinuzzo, and Giuseppe De Nicolao. Bayesian online multitask learning of gaussian processes. *IEEE Transactions on Pattern Analysis and Machine Intelligence*, 32(2):193–205, 2008.

Avishek Saha, Piyush Rai, Hal Daumé, Suresh Venkatasubramanian, et al. Online learning of multiple tasks and their relationships. In *Proceedings of the 14th International Conference on Artificial Intelligence and Statistics*, pp. 643–651, 2011.

Shai Shalev-Shwartz. Online learning: Theory, algorithms, and applications. *PhD thesis, The Hebrew University of Jerusalem, 2007.*, 2007.

Daniel Sheldon. Graphical multi-task learning. Technical report, Computing and Information Science Technical Reports, Cornell University, 2008.

Changjian Shui, Mahdieh Abbasi, Louis-Émile Robitaille, Boyu Wang, and Christian Gagné. A principled approach for learning task similarity in multitask learning. In *Proceedings of the 28th International Joint Conference on Artificial Intelligence*, pp. 3446–3452, 2019.

Suvrit Sra. Fast projections onto mixed-norm balls with applications. *Data Mining and Knowledge Discovery*, 25(2):358–377, 2012.

Jonathan Tuck, Shane Barratt, and Stephen Boyd. A distributed method for fitting Laplacian regularized stratified models. *Journal of Machine Learning Research*, 22(60):1–37, 2021.

Yang Yang, Zhigang Ma, Yi Yang, Feiping Nie, and Heng Tao Shen. Multitask spectral clustering by exploring intertask correlation. *IEEE Transactions on Cybernetics*, 45(5):1083–1094, 2014.

Chi Zhang, Peilin Zhao, Shuji Hao, Yeng Chai Soh, Bu Sung Lee, Chunyan Miao, and Steven CH Hoi. Distributed multi-task classification: a decentralized online learning approach. *Machine Learning*, 107(4): 727–747, 2018.

Yu Zhang and Qiang Yang. A survey on multi-task learning. *IEEE Transactions on Knowledge and Data Engineering*, 2021.

Yu Zhang and Dit Yan Yeung. A convex formulation for learning task relationships in multi-task learning. In *Proceedings of the 26th Conference on Uncertainty in Artificial Intelligence*, pp. 733, 2010.

# A  Technical Proofs

In this Appendix are gathered all the technical proofs of the results stated in the article. First, we provide a notation table recalling the most important notation used along the paper.

| Notation | Meaning |
|---|---|
| $T \in \mathbb{N}$ | Time horizon |
| $N \in \mathbb{N}$ | Number of tasks |
| $d \in \mathbb{N}$ | Dimension of single reference vectors |
| $u_i \in \mathbb{R}^d$, for $i \leq N$ | Reference vector (best model) for task $i$ |
| $\boldsymbol{u} = [u_1, \ldots, u_N] \in \mathbb{R}^{Nd}$ | Compound reference vector |
| $\boldsymbol{u}^{(i)} \in \mathbb{R}^d$ | Block $i$ of the compound vector $\boldsymbol{u}$ ($= u_i$ here) |
| $\bar{\boldsymbol{u}} = (1/N) \sum_i \boldsymbol{u}^{(i)}$ | Average reference vector |
| $\mathrm{Var}_{\|\cdot\|}(\boldsymbol{u}) = (1/(N-1)) \sum_i \|\boldsymbol{u}^{(i)} - \bar{\boldsymbol{u}}\|^2$ | Variance of the reference vectors w.r.t. norm $\|\cdot\|$ |
| $V \subset \mathbb{R}^d$ | Generic set of comparators for a single task |
| $D = \sup_{u \in V} \|u\|$ | Half diameter of $V$ w.r.t. norm $\|\cdot\|$ |
| $\boldsymbol{V} = \{\boldsymbol{u} \in \mathbb{R}^{Nd} : \forall i \leq N, \boldsymbol{u}^{(i)} \in V\}$ | Generic set of compound comparators |
| $\boldsymbol{V}_{\|\cdot\|,\sigma} = \{\boldsymbol{u} \in \boldsymbol{V} : \mathrm{Var}_{\|\cdot\|}(\boldsymbol{u}) \leq \sigma^2 D^2\}$ | Comparators in $\boldsymbol{V}$ with small variance |
| $\boldsymbol{u}_j^{\min} = \min_{i \leq N} \boldsymbol{u}_j^{(i)}$ | Minimum (among tasks) of component $j \leq d$ |
| $\boldsymbol{u}_j^{\max} = \max_{i \leq N} \boldsymbol{u}_j^{(i)}$ | Maximum (among tasks) of component $j \leq d$ |
| $\mathrm{Var}_\Delta(\boldsymbol{u}) = \max_{j \leq d}(1 - \boldsymbol{u}_j^{\min}/\boldsymbol{u}_j^{\max})^2$ | Variance aligned with the probability simplex $\Delta$ |
| $\Delta = \{u \in \mathbb{R}_+^d : \sum_j u_j = 1\}$ | Probability simplex in $\mathbb{R}^d$ |
| $\boldsymbol{\Delta} = \{\boldsymbol{u} \in \mathbb{R}^{Nd} : \forall i \leq N, \boldsymbol{u}^{(i)} \in \Delta\}$ | Compound probability simplex |
| $\boldsymbol{\Delta}_\sigma = \{\boldsymbol{u} \in \boldsymbol{\Delta} : \mathrm{Var}_\Delta(\boldsymbol{u}) \leq \sigma^2\}$ | Comparators in $\boldsymbol{\Delta}$ with small variance |
| $\mathbb{1} \in \mathbb{R}^N$ | Vector filled with 1's |
| $I_N \in \mathbb{R}^{N \times N}$ | Identity matrix of dimension $N$ |
| $A \in \mathbb{R}^{N \times N}$ | Generic interaction matrix |
| $A(b) = (1+b)I_N - (b/N)\mathbb{1}\mathbb{1}^\top$ | Parameterized interaction matrix |
| $\boldsymbol{A} = A \otimes I_d \in \mathbb{R}^{Nd \times Nd}$ | Block version (Kronecker product with $I_d$) of $A$ |
| $\boldsymbol{A}(b) = A(b) \otimes I_d$ | Block version of $A(b)$ |
| $x_{i,t} \in \mathbb{R}^d$, for $i \leq N$ and $t \leq T$ | Prediction maintained by agent $i$ at time $t$ |
| $\boldsymbol{x}_t = [x_{1,t}, \ldots, x_{N,t}] \in \mathbb{R}^{Nd}$ | Compound predictions maintained at time $t$ |
| $i_t \in \mathbb{N}$ | Active agent at time $t$ (chosen by the adversary) |
| $\ell_t : \mathbb{R}^d \to \mathbb{R}$ | Loss function at time $t$ (chosen by the adversary) |
| $g_t = \partial \ell_t(\boldsymbol{x}_t^{(i_t)}) \in \mathbb{R}^d$ | Subgradient of $\ell_t$ at point $\boldsymbol{x}_t^{(i_t)}$ |
| $\boldsymbol{g}_t = [0, \ldots, 0, g_t, 0, \ldots, 0] \in \mathbb{R}^{Nd}$ | Compound subgradient (null outside block $i_t$) |
| $\psi : V \to \mathbb{R}$ | Generic base regularizer |
| $B_\psi : V \times V \to \mathbb{R}$ | Bregman divergence induced by $\psi$ |
| $\boldsymbol{\psi} : \boldsymbol{u} \in \mathbb{R}^{Nd} \mapsto \sum_i \psi(\boldsymbol{u}^{(i)})$ | Compound regularizer |
| $\widetilde{\boldsymbol{\psi}} = \boldsymbol{\psi}(\boldsymbol{A}^{1/2} \cdot \,)$ | Regularizer of interest |

Table 1: Notation Table.

## A.1  Proof of Theorem 1

The first important building block of our proof consists in characterizing the strong convexity of $\widetilde{\boldsymbol{\psi}}$ (Lemma 11). To that end, we need to introduce the following definition and notation about norms.

**Definition 3.** *Let $\mathcal{N}\colon \mathbb{R}^d \to \mathbb{R}$, and $\mathcal{N}'\colon \mathbb{R}^N \to \mathbb{R}$ be two norms. We define the mixed norm $\|\cdot\|_{\mathcal{N},\mathcal{N}'}\colon \mathbb{R}^{Nd} \to \mathbb{R}$ as follows[2]. For all $\boldsymbol{x} \in \mathbb{R}^{Nd}$, $\|\boldsymbol{x}\|_{\mathcal{N},\mathcal{N}'}$ is given by the $\mathcal{N}'$ norm of the $\mathbb{R}^N$ vector composed of the $\mathcal{N}$ norms of the blocks $(\boldsymbol{x}^{(i)})_{i=1}^N$. Formally, it reads*

$$\forall \boldsymbol{x} \in \mathbb{R}^{Nd}, \qquad \|\boldsymbol{x}\|_{\mathcal{N},\mathcal{N}'} = \mathcal{N}'\left(\mathcal{N}(\boldsymbol{x}^{(1)}), \dots, \mathcal{N}(\boldsymbol{x}^{(N)})\right).$$

*When $\mathcal{N} = \|\cdot\|_p$ and $\mathcal{N}' = \|\cdot\|_q$, for $p,q \in [1,+\infty]$, one recovers the standard $\ell_{p,q}$ mixed norm. We use the following shortcut notation*

$$\|\cdot\|_{p,\mathcal{N}'} := \|\cdot\|_{\|\cdot\|_p,\mathcal{N}'} \quad and \quad \|\cdot\|_{\mathcal{N},q} := \|\cdot\|_{\mathcal{N},\|\cdot\|_q}.$$

*Let $s \in \mathbb{N}$, $\|\cdot\|\colon \mathbb{R}^s \to \mathbb{R}$ be any norm, and $M \in \mathbb{R}^{s\times s}$ be a symmetric positive definite matrix. We define the Mahalanobis version of $\|\cdot\|$, denoted $\|\cdot\|_M$, as*

$$\forall x \in \mathbb{R}^s, \qquad \|x\|_M = \|M^{1/2}x\|.$$

*Notice that for $\|\cdot\| = \|\cdot\|_2$, we recover the standard Mahalanobis norm such that $\|x\|_{2,M} = \sqrt{x^\top M x}$. For simplicity, and when it is clear from the context, we use the shortcut notation $\|\cdot\|_M = \|\cdot\|_{2,M}$.*

We are now equipped to establish the strong convexity of $\widetilde{\psi}$.

**Lemma 11.** *Assume that the base regularizer $\psi\colon \mathbb{R}^d \to \mathbb{R}$ is $\lambda$-strongly convex with respect to some norm $\|\cdot\|$. Then, the associated compound regularizer $\widetilde{\boldsymbol{\psi}}$ is $\lambda$-strongly convex with respect to the Mahalanobis mixed norm $\|\cdot\|_{\|\cdot\|,2,\boldsymbol{A}}$.*

*Proof.* First, notice that for every $\boldsymbol{x} \in \mathbb{R}^{Nd}$ it holds

$$\boldsymbol{H}_{\widetilde{\boldsymbol{\psi}}}(\boldsymbol{x}) = \boldsymbol{A}^{1/2}\ \boldsymbol{H}_{\boldsymbol{\psi}}(\boldsymbol{A}^{1/2}\boldsymbol{x})\ \boldsymbol{A}^{1/2},$$

where $\boldsymbol{H}_{\boldsymbol{\psi}}(\boldsymbol{x}) \in \mathbb{R}^{Nd\times Nd}$ denotes the Hessian matrix of regularizer $\boldsymbol{\psi}$ at point $\boldsymbol{x}$. Moreover, due to the definition of the compound regularizer $\boldsymbol{\psi}\colon \boldsymbol{x} \in \mathbb{R}^{Nd} \mapsto \sum_{i=1}^N \psi(\boldsymbol{x}^{(i)})$, matrix $\boldsymbol{H}_{\boldsymbol{\psi}}(\boldsymbol{x})$ is block diagonal and given by

$$\boldsymbol{H}_{\boldsymbol{\psi}}(\boldsymbol{x}) = \begin{pmatrix} H_\psi(\boldsymbol{x}^{(1)}) & 0 & \dots & 0 \\ 0 & H_\psi(\boldsymbol{x}^{(2)}) & \dots & 0 \\ \vdots & \vdots & \ddots & \vdots \\ 0 & \dots & 0 & H_\psi(\boldsymbol{x}^{(N)}) \end{pmatrix}.$$

Assuming that $\psi$ is $\lambda$-strongly convex with respect to norm $\|\cdot\|$, it thus holds for all $\boldsymbol{x}, \boldsymbol{v} \in \mathbb{R}^{Nd}$

$$\begin{aligned} \left\langle \boldsymbol{H}_{\widetilde{\boldsymbol{\psi}}}(\boldsymbol{x})\boldsymbol{v}, \boldsymbol{v} \right\rangle &= \left\langle \boldsymbol{H}_{\boldsymbol{\psi}}(\boldsymbol{A}^{1/2}\boldsymbol{x})(\boldsymbol{A}^{1/2}\boldsymbol{v}), \boldsymbol{A}^{1/2}\boldsymbol{v} \right\rangle \\ &= \sum_{i=1}^N \left\langle H_\psi\left((\boldsymbol{A}^{1/2}\boldsymbol{x})^{(i)}\right)(\boldsymbol{A}^{1/2}\boldsymbol{v})^{(i)}, (\boldsymbol{A}^{1/2}\boldsymbol{v})^{(i)} \right\rangle \\ &\geq \lambda \sum_{i=1}^N \left\| (\boldsymbol{A}^{1/2}\boldsymbol{v})^{(i)} \right\|^2 \\ &= \lambda \|\boldsymbol{v}\|_{\|\cdot\|,2,\boldsymbol{A}}^2. \end{aligned}$$

$\square$

The second important intermediate result (Lemma 12) deals with dual norms, that we recall now.

---

[2]Note that a sufficient condition for $\|\cdot\|_{\mathcal{N},\mathcal{N}'}$ to be a norm is that $\mathcal{N}'(z_1, \dots, z_N) = \mathcal{N}'(|z_1|, \dots, |z_N|)$. In Lemma 11 we use $\mathcal{N}' = \|\cdot\|_2$, which satisfies this property.

**Definition 4.** *Let $s \in \mathbb{N}$, and $\|\cdot\| \colon \mathbb{R}^s \to \mathbb{R}$ be any norm. The* dual norm *of $\|\cdot\|$, denoted $\|\cdot\|_\star$, is defined as*

$$\forall x \in \mathbb{R}^s, \qquad \|x\|_\star = \sup_{y \in \mathbb{R}^s \colon \|y\| \leq 1} \langle x, y \rangle.$$

**Lemma 12.** *Let $p, q \in [1, +\infty]$, and $p^\star, q^\star$ their conjugates such that $1/p + 1/p^\star = 1/q + 1/q^\star = 1$. Then it holds*

$$(\|\cdot\|_p)_\star = \|\cdot\|_{p^\star}, \qquad and \qquad (\|\cdot\|_{p,q})_\star = \|\cdot\|_{p^\star,q^\star}.$$

*Let $s \in \mathbb{N}$, $\|\cdot\| \colon \mathbb{R}^s \to \mathbb{R}$ be any norm, and $M \in \mathbb{R}^{s \times s}$ be a symmetric positive definite matrix. Then it holds*

$$(\|\cdot\|_M)_\star = \|\cdot\|_{\star, M^{-1}}.$$

*Assume furthermore that $s$ can be decomposed into $s = s_1 \times s_2$, for $s_1, s_2 \in \mathbb{N}$. Then, for any norm $\|\cdot\| \colon \mathbb{R}^{s_1} \to \mathbb{R}$ it holds*

$$(\|\cdot\|_{\|\cdot\|,2})_\star = \|\cdot\|_{\|\cdot\|_\star,2}.$$

*In particular, combining the last two results yields:* $\left(\|\cdot\|_{\|\cdot\|,2,\boldsymbol{A}}\right)_\star = \|\cdot\|_{\|\cdot\|_\star,2,\boldsymbol{A}^{-1}}$.

*Proof.* The first result is standard in convex analysis, see *e.g.* Appendix A.1.6 in Boyd et al. (2004). The second result is due to Sra (2012), see Lemma 3 therein. The last two results rely on the following equality (see *e.g.* Example 3.27 in Boyd et al. (2004)). For any norm $\|\cdot\|$, it holds

$$\frac{1}{2}\|\cdot\|_\star^2 = \left(\frac{1}{2}\|\cdot\|^2\right)^\star, \tag{19}$$

where $f^\star$ denotes the Fenchel-Legendre conjugate of $f$, defined as $f^\star(x) = \sup_y \langle x, y \rangle - f(y)$ Bauschke & Combettes (2011). Applying Equation (19) to $\|\cdot\|_M$, we get for all $x \in \mathbb{R}^s$

$$
\begin{aligned}
\left(\frac{1}{2}\left(\|\cdot\|_M\right)_\star^2\right)(x) &= \left(\frac{1}{2}\|\cdot\|_M^2\right)^\star (x) \\
&= \sup_{y \in \mathbb{R}^s} \left\{ \langle x, y \rangle - \frac{1}{2}\|y\|_M^2 \right\} \\
&= \sup_{y \in \mathbb{R}^s} \left\{ \langle M^{-1/2}x, M^{1/2}y \rangle - \frac{1}{2}\left\|M^{1/2}y\right\|^2 \right\} \\
&= \sup_{z \in \mathbb{R}^s} \left\{ \langle M^{-1/2}x, z \rangle - \frac{1}{2}\|z\|^2 \right\} \\
&= \left(\frac{1}{2}\|\cdot\|^2\right)^\star \left(M^{-1/2}x\right) \\
&= \frac{1}{2}\left\|M^{-1/2}x\right\|_\star^2 = \left(\frac{1}{2}\|\cdot\|_{\star,M^{-1}}^2\right)(x).
\end{aligned}
$$

Applying Equation (19) to $\|\cdot\|_{\|\cdot\|,2}$, we get for any $\boldsymbol{x} \in \mathbb{R}^{s_1 \cdot s_2}$

$$
\left(\frac{1}{2}\left(\|\cdot\|_{\|\cdot\|,2}\right)_\star^2\right)(\boldsymbol{x}) = \left(\frac{1}{2}\|\cdot\|_{\|\cdot\|,2}^2\right)^\star(\boldsymbol{x})
$$

$$
= \sup_{\boldsymbol{y}\in\mathbb{R}^{s_1 \cdot s_2}} \left\{\langle\boldsymbol{x},\boldsymbol{y}\rangle - \frac{1}{2}\|\boldsymbol{y}\|_{\|\cdot\|,2}^2\right\}
$$

$$
= \sup_{\boldsymbol{y}\in\mathbb{R}^{s_1 \cdot s_2}} \left\{\sum_{i=1}^{s_2}\langle\boldsymbol{x}^{(i)},\boldsymbol{y}^{(i)}\rangle - \frac{1}{2}\sum_{i=1}^{s_2}\|\boldsymbol{y}^{(i)}\|^2\right\}
$$

$$
= \sum_{i=1}^{s_2} \sup_{\boldsymbol{y}^{(i)}\in\mathbb{R}^{s_1}} \left\{\langle\boldsymbol{x}^{(i)},\boldsymbol{y}^{(i)}\rangle - \frac{1}{2}\|\boldsymbol{y}^{(i)}\|^2\right\}
$$

$$
= \sum_{i=1}^{s_2} \left(\frac{1}{2}\|\cdot\|^2\right)^\star(\boldsymbol{x}^{(i)})
$$

$$
= \frac{1}{2}\sum_{i=1}^{s_2} \|\boldsymbol{x}^{(i)}\|_\star^2 = \left(\frac{1}{2}\|\cdot\|_{\|\cdot\|_\star,2}^2\right)(\boldsymbol{x}).
$$

$\square$

We are now ready to prove Theorem 1. The proof follows from standard arguments to analyze OMD, combined with Lemmas 11 and 12.

**Proof of Theorem 1.** First, notice that the compound representation allows to write

$$
R_T = \sum_{t=1}^T \ell_t(x_t) - \sum_{i=1}^N \inf_{u\in V} \sum_{t:\, i_t=i} \ell_t(u) = \inf_{\boldsymbol{u}\in\boldsymbol{V}} \sum_{t=1}^T \ell_t(x_t) - \ell_t\big(\boldsymbol{u}^{(i_t)}\big).
$$

Next, for any $\boldsymbol{u}\in\boldsymbol{V}$, the convexity and sub-differentiability of $\ell_t$ implies

$$
\sum_{t=1}^T \ell_t(x_t) - \ell_t\big(\boldsymbol{u}^{(i_t)}\big) \le \sum_{t=1}^T \Big\langle g_t, x_t - \boldsymbol{u}^{(i_t)}\Big\rangle = \sum_{t=1}^T \langle\boldsymbol{g}_t, \boldsymbol{x}_t - \boldsymbol{u}\rangle.
$$

Now, observe that update Equation (9) actually defines an OMD on iterate $\boldsymbol{x}_t$, for the sequence of gradients $(\boldsymbol{g}_t)_{t=1}^T$, and with regularizer $\widetilde{\psi}$. Recall also that $\widetilde{\psi}$ is $\lambda$-strongly convex with respect to $\|\cdot\|_{\|\cdot\|,2,\boldsymbol{A}}$ (Lemma 11), whose dual norm is $\|\cdot\|_{\|\cdot\|_\star,2,\boldsymbol{A}^{-1}}$ (Lemma 12). Applying the standard OMD bound of Equation (3), we thus obtain that for all $\boldsymbol{u}\in\boldsymbol{V}$ and $\eta>0$ it holds

$$
\sum_{t=1}^T \langle\boldsymbol{g}_t, \boldsymbol{x}_t - \boldsymbol{u}\rangle \le \frac{B_{\widetilde{\psi}}(\boldsymbol{u},\boldsymbol{x}_1)}{\eta} + \frac{\eta}{2\lambda}\sum_{t=1}^T \|\boldsymbol{g}_t\|_{\|\cdot\|_\star,2,\boldsymbol{A}^{-1}}^2.
$$

Then, we use that for all $\boldsymbol{x},\boldsymbol{y}\in\mathbb{R}^{Nd}$ it holds $B_{\widetilde{\psi}}(\boldsymbol{x},\boldsymbol{y}) = B_{\psi}\big(\boldsymbol{A}^{1/2}\boldsymbol{x}, \boldsymbol{A}^{1/2}\boldsymbol{y}\big)$, and that

$$
\|\boldsymbol{g}_t\|_{\|\cdot\|_\star,2,\boldsymbol{A}^{-1}}^2 = \left\|\boldsymbol{A}^{-1/2}\boldsymbol{g}_t\right\|_{\|\cdot\|_\star,2}^2
$$

$$
= \sum_{i=1}^N \left\|A_{ii_t}^{-1/2} g_t\right\|_\star^2
$$

$$
= \sum_{i=1}^N \left(A_{ii_t}^{-1/2}\right)^2 \|g_t\|_\star^2
$$

$$
= A_{i_t i_t}^{-1} \|g_t\|_\star^2
$$

$$
\le \max_{i\le N} A_{ii}^{-1} \|g_t\|_\star^2.
$$

Combining all arguments, we finally obtain

$$R_T(\boldsymbol{u}) \leq \frac{B_{\boldsymbol{\psi}}\big(\boldsymbol{A}^{1/2}\boldsymbol{u}, \boldsymbol{A}^{1/2}\boldsymbol{x}_1\big)}{\eta} + \max_{i \leq N} A_{ii}^{-1} \frac{\eta}{2\lambda} \sum_{t=1}^{T} \|g_t\|_{\star}^2 .$$

The second claim of the theorem is obtained in a similar fashion. Standard OMD results (Orabona, 2019, Theorem 6.8) give that

$$R_T(\boldsymbol{u}) \leq \max_{t \leq T} \frac{B_{\widetilde{\boldsymbol{\psi}}}(\boldsymbol{u}, \boldsymbol{x}_t)}{\eta_T} + \frac{1}{2\lambda} \sum_{t=1}^{T} \eta_t \|\boldsymbol{g}_t\|_{\|\cdot\|_{\star},2,\boldsymbol{A}^{-1}}^2 .$$

Replacing $B_{\widetilde{\boldsymbol{\psi}}}(\boldsymbol{u}, \boldsymbol{x}_t)$ and $\|\boldsymbol{g}_t\|_{\|\cdot\|_{\star},2,\boldsymbol{A}^{-1}}^2$ with $B_{\boldsymbol{\psi}}\big(\boldsymbol{A}^{1/2}\boldsymbol{u}, \boldsymbol{A}^{1/2}\boldsymbol{x}_t\big)$ and $\max_{i \leq N} A_{ii}^{-1} \|g_t\|_{\star}^2$ respectively yields the desired result. $\square$

## A.2 Proof of Proposition 2

First, it is easy to check that we have

$$A(b) = (1+b)I_N - b\frac{\mathbb{1}\mathbb{1}^\top}{N}$$

$$A(b)^{1/2} = \sqrt{1+b}\, I_N + (1 - \sqrt{1+b})\frac{\mathbb{1}\mathbb{1}^\top}{N}$$

$$A(b)^{-1} = \frac{1}{1+b}I_N + \frac{b}{(1+b)}\frac{\mathbb{1}\mathbb{1}^\top}{N}.$$

This implies

$$\max_{i \leq N} \, [A(b)^{-1}]_{ii} = \frac{b+N}{(1+b)N}, \tag{20}$$

and

$$
\begin{aligned}
2B_{\boldsymbol{\psi}}\Big(\boldsymbol{A}(b)^{1/2}\boldsymbol{u}, \boldsymbol{A}(b)^{1/2}\boldsymbol{0}\Big) &= \sum_{i=1}^{N} \left\|\big(\boldsymbol{A}(b)^{1/2}\boldsymbol{u}\big)^{(i)}\right\|_2^2 \\
&= \sum_{i=1}^{N} \left\|\sqrt{1+b}\,\boldsymbol{u}^{(i)} + (1-\sqrt{1+b})\bar{\boldsymbol{u}}\right\|_2^2 \\
&= \sum_{i=1}^{N} \left\|\sqrt{1+b}\,(\boldsymbol{u}^{(i)} - \bar{\boldsymbol{u}}) + \bar{\boldsymbol{u}}\right\|_2^2 \\
&= (1+b)\sum_{i=1}^{N} \left\|\boldsymbol{u}^{(i)} - \bar{\boldsymbol{u}}\right\|_2^2 + \sum_{i=1}^{N} \|\bar{\boldsymbol{u}}\|_2^2 + 2\sqrt{1+b}\sum_{i=1}^{N} \big\langle \boldsymbol{u}^{(i)} - \bar{\boldsymbol{u}}, \bar{\boldsymbol{u}}\big\rangle \\
&= \sum_{i=1}^{N} \Big(\left\|\boldsymbol{u}^{(i)} - \bar{\boldsymbol{u}}\right\|_2^2 + \|\bar{\boldsymbol{u}}\|_2^2\Big) + b(N-1)\mathrm{Var}_{\|\cdot\|_2}(\boldsymbol{u}) \\
&= \sum_{i=1}^{N} \left\|\boldsymbol{u}^{(i)}\right\|_2^2 + b(N-1)\mathrm{Var}_{\|\cdot\|_2}(\boldsymbol{u}) . \tag{21}
\end{aligned}
$$

Substituting Equation (20), Equation (21) into Equation (11), and using the definition of $\boldsymbol{V}_{\|\cdot\|_2,\sigma}$, we get that $R_T(\boldsymbol{u})$ is smaller than

$$\frac{\|\boldsymbol{u}\|_2^2 + b(N-1)\mathrm{Var}_{\|\cdot\|_2}(\boldsymbol{u})}{2\eta} + \frac{\eta T L_g^2}{2} \frac{b+N}{(1+b)N} \quad \leq \quad \frac{ND^2(1+b\frac{N-1}{N}\sigma^2)}{2\eta} + \frac{\eta T L_g^2}{2} \frac{b+N}{(1+b)N} .$$

Setting $\eta = \frac{ND}{L_g}\sqrt{\frac{\left(1+b\frac{N-1}{N}\sigma^2\right)(1+b)}{(b+N)T}}$, we get

$$R_T \leq DL_g \sqrt{\frac{\left(1+b\frac{N-1}{N}\sigma^2\right)(b+N)}{1+b}T}\,.$$

We now optimize on $b$. Let $\alpha = \frac{N-1}{N}\sigma^2$, the optimality condition writes

$$(1+\alpha N + 2\alpha b^*)(1+b^*) = (1+\alpha b^*)(b^*+N)$$

$$1 + (\alpha-1)N + 2\alpha b^* + \alpha b^{*2} = 0$$

$$b^* = \sqrt{\frac{1-\alpha}{\alpha}(N-1)} - 1 = \sqrt{1 + \frac{1-\sigma^2}{\sigma^2}N} - 1\,.$$

And we have

$$\frac{(1+\alpha b^*)(b^*+N)}{1+b^*} = 1 + \alpha N + 2\alpha b^*$$

$$= 1 + \sigma^2(N-1) + 2\sigma^2\frac{N-1}{N}\left(\sqrt{1 + \frac{1-\sigma^2}{\sigma^2}N} - 1\right)\,.$$

Hence, the bound becomes $DL_g\sqrt{F(\sigma)\,T}$, with

$$F(\sigma) = 1 + \sigma^2(N-1) + 2\sigma^2\frac{N-1}{N}\left(\sqrt{1 + \frac{1-\sigma^2}{\sigma^2}N} - 1\right)\,.$$

Note that

$$2\alpha b^* \leq 2\sqrt{\alpha(1-\alpha)(N-1)} \leq \alpha(1-\alpha)(N-1) + 1 \leq 1 + \alpha N\,,$$

so that we have $F(\sigma) \leq 2(1+\sigma^2(N-1))$, and the bound becomes $DL_g\sqrt{1+\sigma^2(N-1)}\sqrt{2T}$, a value which can also be achieved directly by setting $b = N$ above. □

### A.3 Proof of Proposition 3

This lower bound is proved by adapting the standard lower bound for (single-agent) Online Linear Optimization, see, e.g., (Orabona, 2019, Chapter 5). We consider linear losses, i.e., we assume that for all $t \leq T$ there exists $g_t \in \mathbb{R}^d$ such that $\ell_t(x) = \langle g_t, x\rangle$. Our goal is to show that for any any sequence $\boldsymbol{x}_1, \ldots, \boldsymbol{x}_T$ we can construct a sequence of losses (or equivalently of $g_t$) such that the regret is lower bounded. Recall that $d$ is the dimension of the set $\boldsymbol{V}$ and $D > 0$ its radius, and let $\sigma < 1$. Assume that $N$ is even, smaller than $2d$, and for all $i \in \{1, \ldots, N/2\}$, let

$$\boldsymbol{w}^{(2i-1)} = -\sqrt{(N-1)/N}D\sigma e_i \qquad \text{and} \qquad \boldsymbol{w}^{(2i)} = \sqrt{(N-1)/N}D\sigma e_i\,,$$

where $(e_i)_{i \leq d}$ is the canonical basis of $\mathbb{R}^d$. It is easy to check that $\|\boldsymbol{w}^{(i)}\|_2 \leq D$, and that $\text{Var}(\boldsymbol{w}) = \sigma^2 D^2$, such that $\boldsymbol{w} \in \boldsymbol{V}_{\|\cdot\|_2, \sigma}$. Assume that $N$ divides $T$, and that the agents are activated in a cyclic fashion, i.e., $i_t = 1 + t \mod N$. We introduce the family of gradient vectors

$$g_t = \epsilon_{\lceil t/N\rceil}\frac{L_g\boldsymbol{w}^{(i_t)}}{\|\boldsymbol{w}^{(i_t)}\|_2}\,, \tag{22}$$

where $\epsilon_1, \ldots, \epsilon_{T/N}$ are valued in $\{-1, 1\}$, with exact values to be determined later on. Indeed, we show that for any sequence of predictions, there exists a choice of $\epsilon_1, \ldots, \epsilon_{T/N}$ such that the regret is lower bounded. To that end, we use the fact that for any function $F: \{-1, 1\}^{T/N} \to \mathbb{R}$, and any probability distribution $P$ with support in $\{-1, 1\}$, we have

$$\sup_{\boldsymbol{\epsilon} \in \{-1,1\}^{T/N}} F(\boldsymbol{\epsilon}) \geq \mathbb{E}_{\boldsymbol{\epsilon} \sim P^{\otimes T/N}}[F(\boldsymbol{\epsilon})]\,.$$

In particular, we choose $P$ to be the Rademacher distribution, such that $\mathbb{P}\{\epsilon = -1\} = \mathbb{P}\{\epsilon = 1\} = 1/2$, and all expectations are now understood to be taken with respect to this distribution. Note that we have $\|g_t\|_2 \le L_g$ and $\mathbb{E}_\epsilon[g_t] = 0$, for all $t$. Given any sequence $\boldsymbol{x}_1, \ldots, \boldsymbol{x}_T$, we can use gradients Equation (22) and obtain

$$
\sup_{\epsilon_1, \ldots, \epsilon_{T/N}} R_T \ge \mathbb{E}_\epsilon \left[ \sum_{t=1}^T \left\langle g_t, \boldsymbol{x}^{(i_t)} \right\rangle - \min_{\boldsymbol{u} \in \boldsymbol{V}_{\|\cdot\|_2, \sigma}} \sum_{t=1}^T \left\langle g_t, \boldsymbol{u}^{(i_t)} \right\rangle \right]
$$

$$
= \mathbb{E}_\epsilon \left[ \max_{\boldsymbol{u} \in \boldsymbol{V}_{\|\cdot\|_2, \sigma}} \sum_{t=1}^T \epsilon_{\lceil t/N \rceil} \frac{L_g}{\|\boldsymbol{w}^{(i_t)}\|_2} \left\langle \boldsymbol{w}^{(i_t)}, \boldsymbol{u}^{(i_t)} \right\rangle \right]
$$

$$
= \frac{L_g \sqrt{N}}{D\sigma\sqrt{N-1}} \mathbb{E}_\epsilon \left[ \max_{\boldsymbol{u} \in \boldsymbol{V}_{\|\cdot\|_2, \sigma}} \sum_{t=1}^T \epsilon_{\lceil t/N \rceil} \left\langle \boldsymbol{w}^{(i_t)}, \boldsymbol{u}^{(i_t)} \right\rangle \right]
$$

$$
= \frac{L_g \sqrt{N}}{D\sigma\sqrt{N-1}} \mathbb{E}_\epsilon \left[ \max_{\boldsymbol{u} \in \boldsymbol{V}_{\|\cdot\|_2, \sigma}} \sum_{\tau=1}^{T/N} \epsilon_\tau \left\langle \boldsymbol{w}, \boldsymbol{u} \right\rangle \right]
$$

$$
\ge \frac{L_g \sqrt{N}}{D\sigma\sqrt{N-1}} \mathbb{E}_\epsilon \left[ \max_{\boldsymbol{u} \in \{-\boldsymbol{w}, \boldsymbol{w}\}} \sum_{\tau=1}^{T/N} \epsilon_\tau \left\langle \boldsymbol{w}, \boldsymbol{u} \right\rangle \right]
$$

$$
\ge D L_g \sigma \sqrt{N(N-1)} \, \mathbb{E}_\epsilon \left| \sum_{\tau=1}^{T/N} \epsilon_\tau \right| \tag{23}
$$

$$
\ge \frac{D L_g}{2} \sigma \sqrt{N-1} \sqrt{2T} \,, \tag{24}
$$

where in Equation (23) we used $\|\boldsymbol{w}\|_2^2 = (N-1)D^2\sigma^2$ and in Equation (24) we used the Khintchine inequality, see, e.g., (Cesa-Bianchi & Lugosi, 2006, Lemma 8.2). We now combine this lower bound with the one obtained by choosing $i_t = 1$ for all $t$ and applying the standard single-agent lower bound $\frac{DL_g}{2}\sqrt{2T}$, see, e.g., (Orabona, 2019, Theorem 5.1). Combining these two bounds, we obtain that the regret is lower bounded by

$$
\frac{DL_g}{2}\sqrt{2T} \max\left\{1, \sigma\sqrt{N-1}\right\} \ge \frac{DL_g}{4}\sqrt{2T}\left(1 + \sigma\sqrt{N-1}\right) \ge \frac{1}{4}\left(DL_g\sqrt{1+\sigma^2(N-1)}\sqrt{2T}\right),
$$

which is only $1/4$ of the upper bound Equation (14). Hence, with knowledge of $\sigma^2$, there is no algorithm with better multitask acceleration than MT-OMD, up to constant factors. $\qquad\square$

### A.4 Proof of Proposition 4

Consider the following setting. Let $N = 2d$, $\sigma < 1$, $u_0 \in \mathbb{R}^d$ be such that $\|u_0\|_2^2 = 1 - \sigma^2$, and set for all $i \le d$:

$$
\boldsymbol{u}^{(2i-1)} = u_0 - \sqrt{(N-1)/N}\,\sigma e_i\,, \qquad \text{and} \qquad \boldsymbol{u}^{(2i)} = u_0 + \sqrt{(N-1)/N}\,\sigma e_i\,,
$$

where $(e_i)_{i \le d}$ is the canonical basis of $\mathbb{R}^d$. It is easy to check that $\left\|\boldsymbol{u}^{(i)}\right\|_2^2 \ge 1 - 2\sigma^2$ for all $i \le N$, and that $\mathrm{Var}_{\|\cdot\|_2}(\boldsymbol{u}) = \sigma^2$. Then, a standard lower bound for OGD (Orabona, 2019, Theorem 5.1) applied to the $N$ individual independent OGDs of IT-OGD with linear losses, unit-norm loss gradients, and cyclic activations such that $T_i = T/N$, gives that

$$
R_T^{\text{IT-OGD}} \ge \sum_{i=1}^N \left\|\boldsymbol{u}^{(i)}\right\|_2 \frac{\sqrt{2T_i}}{4} \ge \frac{\sqrt{2(1-2\sigma^2)NT}}{4}\,.
$$

This lower bound is strictly greater than the MT-OGD upper bound Equation (14) as soon as $\frac{\sqrt{(1-2\sigma^2)N}}{4} > \sqrt{1 + \sigma^2(N-1)}$, or equivalently as soon as $\sigma^2 \le \frac{N-16}{18N-16}$. $\qquad\square$

## A.5 Proof of Proposition 5

As discussed in the main body, the analysis for general norms is made more complex by the fact that Equation (21) does not hold with equality anymore. Instead, a series of approximations leveraging the properties of norms is required. Indeed, for any norm $\|\cdot\|$, and regularizer $\psi = \frac{1}{2}\|\cdot\|^2$, it holds

$$
\begin{aligned}
2B_{\psi}\left(\boldsymbol{A}(b)^{1/2}\boldsymbol{u}, \boldsymbol{A}(b)^{1/2}\boldsymbol{0}\right) &= \sum_{i=1}^{N}\left\|\left(\boldsymbol{A}(b)^{1/2}\boldsymbol{u}\right)^{(i)}\right\|^2 \\
&= \sum_{i=1}^{N}\left\|\sqrt{1+b}\,\boldsymbol{u}^{(i)} + (1-\sqrt{1+b})\bar{\boldsymbol{u}}\right\|^2 \\
&= \sum_{i=1}^{N}\left\|\bar{\boldsymbol{u}} + \sqrt{1+b}\,(\boldsymbol{u}^{(i)} - \bar{\boldsymbol{u}})\right\|^2 \\
&\leq 2\left(N\|\bar{\boldsymbol{u}}\|^2 + (1+b)\sum_{i=1}^{N}\left\|\boldsymbol{u}^{(i)} - \bar{\boldsymbol{u}}\right\|^2\right) \\
&\leq 2\left(2ND^2 + b(N-1)\mathrm{Var}_{\|\cdot\|}(\boldsymbol{u})\right) \\
&\leq 4ND^2\left(1 + b\frac{N-1}{N}\sigma^2\right).
\end{aligned}
\tag{25}
$$

In comparison to Equation (21), the bound is thus multiplied by 4. The rest of the proof (i.e. the optimization in $\eta$ and $b$) is similar to that of Proposition 2, and we obtain that for all $\boldsymbol{u} \in \boldsymbol{V}_{\|\cdot\|,\sigma}$ it holds:

$$
R_T(\boldsymbol{u}) \leq DL_g\sqrt{1 + \sigma^2(N-1)}\sqrt{8T}.
$$

An interesting use case unlocked by the previous result is the use of the $p$-norm (for $p \in [1,2]$) on the the probability simplex $\Delta = \{x \in \mathbb{R}_+^d : \sum_j x_j = 1\}$. Indeed, by a careful tuning of $p$ one can derive bounds scaling as $\sqrt{\ln d}$, instead of $d$ for OGD. Interestingly, this improvement is orthogonal to our multitask acceleration, so that it is possible to benefit from both. Recall that regularizer $\frac{1}{2}\|\cdot\|_p^2$ is $(p-1)$ strongly convex with respect to $\|\cdot\|_p$ (Shalev-Shwartz, 2007, Lemma 17), and that the dual norm of $\|\cdot\|_p$ is $\|\cdot\|_q$, with $q$ such that $1/p + 1/q = 1$. Consider $V = \Delta$, and loss functions such that $\|g_t\|_\infty \leq L_\infty$ for all $g_t \in \partial\ell_t(x)$, $x \in \Delta$. One can check that $\|g_t\|_q^2 \leq L_\infty^2 d^{2/q}$. Then, substituting Equation (25) into Equation (11) and using the previous remark, we get that for all $\boldsymbol{u} \in \boldsymbol{\Delta}_{\|\cdot\|_p,\sigma}$ it holds

$$
\begin{aligned}
R_T(\boldsymbol{u}) &\leq \frac{2ND^2\left(1 + b\frac{N-1}{N}\sigma^2\right)}{\eta} + \frac{b+N}{(1+b)N}\frac{\eta}{2(p-1)}TL_\infty^2 d^{2/q} \\
&\leq 2DL_\infty\frac{d^{1/q}}{\sqrt{p-1}}\sqrt{\frac{\left(1 + b\frac{N-1}{N}\sigma^2\right)(b+N)}{1+b}T},
\end{aligned}
$$

with the choice $\eta = 2ND\sqrt{\frac{(1+b)\left(1+b\frac{N-1}{N}\sigma^2\right)}{b+N}\frac{(p-1)}{TL_\infty^2 d^{2/q}}}$. Hence, the bound obtained is the product of two terms, one depending only on $b$, the other only on $p$. We can thus optimize independently. The term in $b$ is the same as in previous proofs, we can reuse the analysis made for Proposition 2. The term in $d$ is the same as in the original proof Grove et al. (1997); Gentile (2003), and the optimization is thus similar. We repeat it here for completeness. One has $d^{1/q}/\sqrt{p-1} = \sqrt{q-1}d^{1/q} \leq \sqrt{q}d^{1/q}$. By differentiating with respect to $q$, the last term is minimized for $q = 2\ln d$, with a value of $\sqrt{2e\ln d}$. The final bound obtained is

$$
L_\infty\sqrt{1 + \sigma^2(N-1)}\sqrt{16e\,T\ln d}.
$$

We conclude with a few remarks. First, in order to ensure that $q \geq 2$ (we need $p \leq 2$), we may assume that $d \geq 3$. Second, note that the improvement on the dependence with respect to $d$ comes at the price of a stronger variance condition, as we have $\mathrm{Var}_{\|\cdot\|_2}(\boldsymbol{u}) \leq \mathrm{Var}_{\|\cdot\|_p}(\boldsymbol{u})$ for all $p \leq 2$. If one is interested in a condition independent from $d$ (indeed $p$ depends on $q$, which depends on $d$), the variance with respect to $\|\cdot\|_1$ can be used. Finally, note that we have used $D = \sup_{x\in\Delta}\|x\|_p \leq \sup_{x\in\Delta}\|x\|_1 = 1$. $\qquad\square$

### A.6 Proof of Proposition 6

This proposition builds upon the second claim of Theorem 1. We start by detailing the proof for the specific choice $\|\cdot\| = \|\cdot\|_2$. Observe that for $\psi = \frac{1}{2}\|\cdot\|_2^2$ and all $\boldsymbol{u}, \boldsymbol{x} \in \boldsymbol{V}_{\|\cdot\|_2,\sigma}$ we have

$$B_{\psi}\big(\boldsymbol{A}^{1/2}\boldsymbol{u}, \boldsymbol{A}^{1/2}\boldsymbol{x}\big) = \frac{1}{2}\|\boldsymbol{u} - \boldsymbol{x}\|_{\boldsymbol{A}}^2 \leq \|\boldsymbol{u}\|_{\boldsymbol{A}}^2 + \|\boldsymbol{x}\|_{\boldsymbol{A}}^2 \leq 2ND^2\left(1 + b\frac{N-1}{N}\sigma^2\right).$$

Substituting into Equation (12), we obtain

$$R_T(\boldsymbol{u}) \leq \frac{2ND^2\left(1 + b\frac{N-1}{N}\sigma^2\right)}{\eta_T} + \frac{b+N}{2(1+b)N}\sum_{t=1}^T \eta_t\|g_t\|_2^2$$

$$= \frac{b+N}{(1+b)N}\left(\frac{\bar{D}^2}{2\eta_T} + \frac{1}{2}\sum_{t=1}^T \eta_t\|g_t\|_2^2\right),$$

with $\bar{D}^2 = \frac{4N^2D^2(1+b)\left(1+b\frac{N-1}{N}\sigma^2\right)}{b+N}$. Using $\eta_t = \frac{\sqrt{2}\bar{D}}{2\sqrt{\sum_{i=1}^t \|g_i\|_2^2}} = ND\sqrt{\frac{2(1+b)\left(1+b\frac{N-1}{N}\sigma^2\right)}{(b+N)\sum_{i=1}^t \|g_i\|_2^2}}$, (Orabona, 2019,

Lemma 4.13) gives

$$R_T(\boldsymbol{u}) \leq \frac{b+N}{(1+b)N}\sqrt{2}\bar{D}\sqrt{\sum_{t=1}^T \|g_t\|_2^2} \leq D\sqrt{\frac{(b+N)\left(1 + b\frac{N-1}{N}\sigma^2\right)}{1+b}}\sqrt{8\sum_{t=1}^T \|g_t\|_2^2}. \tag{26}$$

Choosing $b = N$ concludes the proof. In comparison to Proposition 2 (and assuming that $\|g_t\|_2 \leq L_g$), the bound is multiplied by $\sqrt{8}$. One $\sqrt{2}$ multiplication is due to the choice of $\eta_t$, while the other $\sqrt{4}$ multiplication comes from the upper bound on $\max_{t \leq T} B_{\psi}\big(\boldsymbol{A}^{1/2}\boldsymbol{u}, \boldsymbol{A}^{1/2}\boldsymbol{x}_t\big)$, which is 4 times bigger than the upper bound of $B_{\psi}\big(\boldsymbol{A}^{1/2}\boldsymbol{u}, \boldsymbol{A}^{1/2}\boldsymbol{x}_1\big)$. The proof for any norm follows the same path. The only difference is on bounding $\max_{t \leq T} B_{\psi}\big(\boldsymbol{A}^{1/2}\boldsymbol{u}, \boldsymbol{A}^{1/2}\boldsymbol{x}_t\big)$, which is 4 times bigger than the same quantity for the Euclidean case, see Equation (25). Therefore, an additional $\sqrt{4} = 2$ factor is added. □

### A.7 Proof of Proposition 7

Using Equation (26) with $b = N$, the smoothness of the losses gives

$$R_T(\boldsymbol{u}) \leq 4D\sqrt{1 + \sigma^2(N-1)}\sqrt{M\sum_{t=1}^T \ell_t\big(\boldsymbol{x}_t^{(i_t)}\big)}.$$

Using Lemma 4.20 in Orabona (2019), we obtain

$$R_T(\boldsymbol{u}) \leq 8D\sqrt{1 + \sigma^2(N-1)}\left(2MD\sqrt{1 + \sigma^2(N-1)} + \sqrt{M\sum_{t=1}^T \ell_t\big(\boldsymbol{u}^{(i_t)}\big)}\right).$$

As for Proposition 6, the claim for general losses is obtained by multiplying the bound by 2. □

### A.8 Proof of Proposition 8

Let $\boldsymbol{x}_1 = [x^*, \ldots, x^*]$, and observe that $\boldsymbol{A}(b)^{1/2}\boldsymbol{x}_1 = \boldsymbol{x}_1$. Then, for all $\boldsymbol{u} \in \boldsymbol{\Delta}$ such that $\boldsymbol{A}(b)^{1/2}\boldsymbol{u} \in \boldsymbol{\Delta}$ we have

$$B_{\psi}\big(\boldsymbol{A}(b)^{1/2}\boldsymbol{u}, \boldsymbol{A}(b)^{1/2}\boldsymbol{x}_1\big) = B_{\psi}\big(\boldsymbol{A}(b)^{1/2}\boldsymbol{u}, \boldsymbol{x}_1\big) = \sum_{i=1}^N B_{\psi}\big(\big(\boldsymbol{A}(b)^{1/2}\boldsymbol{u}\big)^{(i)}, x^*\big) \leq NC. \tag{27}$$

Plugging Equation (27) into Equation (11), we obtain for all $\boldsymbol{u} \in \boldsymbol{\Delta}$:

$$R_T(\boldsymbol{u}) \leq \frac{NC}{\eta} + \frac{\eta(b+N)}{2\lambda(1+b)N}TL_g^2 \leq L_g\sqrt{\frac{2}{\lambda}\frac{b+N}{b+1}CT}, \tag{28}$$

where we have set $\eta = \frac{N}{L_g}\sqrt{\frac{2\lambda(1+b)C}{(b+N)T}}$. The next natural question is *how to choose b?* As bound Equation (28) is decreasing in $b$, one is encouraged to choose $b$ as large as possible. However, recall that Equation (27) requires $\boldsymbol{A}(b)^{1/2}\boldsymbol{u}$ to be in $\boldsymbol{\Delta}$. So the optimal choice is $b^* = \max\{b \geq 0\colon \boldsymbol{A}(b)^{1/2}\boldsymbol{u} \in \boldsymbol{\Delta}\}$. This value unfortunately depends on $\boldsymbol{u}$ and cannot be used uniformly over $\boldsymbol{\Delta}$. However, the variance condition for $\boldsymbol{\Delta}_\sigma$ allows to choose a global $b$, as we show now. Let $\boldsymbol{u} \in \boldsymbol{\Delta}_\sigma$. Recall that for all $i \leq N$ we have

$$\left(\boldsymbol{A}(b)^{1/2}\boldsymbol{u}\right)^{(i)} = \sqrt{1+b}\,\boldsymbol{u}^{(i)} + (1 - \sqrt{1+b})\bar{\boldsymbol{u}}\,.$$

We have to check that these vectors are in the simplex for all $i \leq N$. There are two conditions that a vector $x$ should verify to be in the simplex

$$(i)\;\; \mathbb{1}^\top x = 1, \qquad \text{and} \qquad (ii)\;\; x_j \geq 0 \quad \forall j \leq d.$$

It is immediate to see that the first condition is always satisfied. To analyze the second condition, we recall the following notation from Definition 2

$$\forall j \leq d, \qquad \boldsymbol{u}_j^{\max} = \max_{i \leq N}\,\boldsymbol{u}_j^{(i)}, \qquad \boldsymbol{u}_j^{\min} = \min_{i \leq N}\,\boldsymbol{u}_j^{(i)}\,.$$

Now, let $j \leq d$. A sufficient condition for $(ii)$ to hold for every $\left(\boldsymbol{A}(b)\boldsymbol{u}\right)^{(i)}$, $i \leq N$, is

$$0 \leq \sqrt{1+b}\,\boldsymbol{u}_j^{\min} + (1 - \sqrt{1+b})\boldsymbol{u}_j^{\max}\,,$$

Or, equivalently,

$$b \leq \left(\frac{\boldsymbol{u}_j^{\max}}{\boldsymbol{u}_j^{\max} - \boldsymbol{u}_j^{\min}}\right)^2 - 1\,.$$

Since this condition must hold for every $j \leq d$, the overall condition is

$$b \;\leq\; \min_{j \leq d}\left(\frac{\boldsymbol{u}_j^{\max}}{\boldsymbol{u}_j^{\max} - \boldsymbol{u}_j^{\min}}\right)^2 - 1 = \frac{1}{\sigma^2} - 1 = \frac{1-\sigma^2}{\sigma^2}\,.$$

This condition is quite intuitive. If all best models are equal, $\boldsymbol{u}_j^{\min} = \boldsymbol{u}_j^{\max}$ for all $j \leq d$, one can choose $b = +\infty$, and achieves a bound independent from $N$. On the contrary, if there exists $j \leq d$ such that $\boldsymbol{u}_j^{\max} = 1$ and $\boldsymbol{u}_j^{\min} = 0$, i.e., two different corners are linked, then $b = 0$ is the only possible choice, and a $\sqrt{N}$ dependence is unavoidable. Finally, with $b = (1 - \sigma^2)/\sigma^2 \geq 0$, bound Equation (28) becomes

$$L_g\sqrt{1 + \sigma^2(N-1)}\sqrt{2CT/\lambda}\,.$$

$\square$

## A.9 Proof of Theorem 9

Let $\varepsilon > 0$, and consider the covering $\mathcal{C}_\varepsilon = \{\varepsilon, 2\varepsilon, \dots, 1\}$, of cardinality $1/\varepsilon$. Let $\boldsymbol{u} \in \boldsymbol{V}$, we want to derive an upper bound of the regret achieved by the best expert in $\mathcal{C}_\varepsilon$ against $\boldsymbol{u}$. First, assume that $\text{Var}_{\|\cdot\|_2}(\boldsymbol{u}) \leq 1$, and let $\bar{\sigma}^2 = \inf\{z \in \mathcal{C}_\varepsilon\colon \text{Var}_{\|\cdot\|_2}(\boldsymbol{u}) \leq z\}$. Note that by definition we have $\text{Var}_{\|\cdot\|_2}(\boldsymbol{u}) \leq \bar{\sigma}^2 \leq \text{Var}_{\|\cdot\|_2}(\boldsymbol{u}) + \varepsilon$. Now, let us bound the regret of MT-OGD run with $\sigma^2 = \bar{\sigma}^2$. Recall that for any fixed learning rate $\eta$, the regret of MT-OGD is bounded by

$$\frac{ND^2\left(1 + (N-1)\text{Var}_{\|\cdot\|_2}(\boldsymbol{u})\right)}{2\eta} + \eta\frac{TL_g^2}{N} \leq \frac{ND^2\left(1 + (N-1)\bar{\sigma}^2\right)}{2\eta} + \eta\frac{TL_g^2}{N}\,. \tag{29}$$

MT-OGD run with $\sigma^2 = \bar{\sigma}^2$ uses $\eta = \frac{ND}{L_g}\sqrt{\frac{1+(N-1)\bar{\sigma}^2}{2T}}$. Plugging this into Equation (29), we get that the regret is upper bounded by

$$
\begin{aligned}
DL_g\sqrt{1+\bar{\sigma}^2(N-1)}\sqrt{2T} &\leq DL_g\left(1+\sqrt{\text{Var}_{\|\cdot\|_2}(\boldsymbol{u})\cdot N}+\sqrt{\varepsilon N}\right)\sqrt{2T} \\
&= DL_g\left(1+\sqrt{\min\left\{\text{Var}_{\|\cdot\|_2}(\boldsymbol{u}),1\right\}\cdot N}+\sqrt{\varepsilon N}\right)\sqrt{2T}.
\end{aligned}
\tag{30}
$$

On the other hand, if $\text{Var}_{\|\cdot\|_2}(\boldsymbol{u}) \geq 1$, we know that MT-OGD run with $\sigma^2 = 1$ is equivalent to independent OGDs and has a regret bounded by

$$
\begin{aligned}
DL_g\sqrt{NT} &\leq DL_g\left(1+\sqrt{N}+\sqrt{\varepsilon N}\right)\sqrt{2T} \\
&= DL_g\left(1+\sqrt{\min\left\{\text{Var}_{\|\cdot\|_2}(\boldsymbol{u}),1\right\}\cdot N}+\sqrt{\varepsilon N}\right)\sqrt{2T}.
\end{aligned}
\tag{31}
$$

Combining Equation (30) and Equation (31), we know that in any case the best expert in $\mathcal{C}_\varepsilon$ has always a regret against $\boldsymbol{u}$ smaller than

$$
DL_g\left(1+\sqrt{\min\left\{\text{Var}_{\|\cdot\|_2}(\boldsymbol{u}),1\right\}\cdot N}+\sqrt{\varepsilon N}\right)\sqrt{2T}.
\tag{32}
$$

Let us now compute the regret of Hedge-MT-OGD against the best expert in $\mathcal{C}_\varepsilon$. By the analysis of Hedge with linear combination of the experts, we know that it is bounded by (Orabona, 2019):

$$
\frac{\sqrt{2}}{2}L_\infty\sqrt{T\log(1/\varepsilon)},
\tag{33}
$$

where $L_\infty$ is an upper bound of the infinity norm of the gradients received by Hedge. The latter are equal to the vectors of losses incurred by the different experts at each time step. With linear(ized) losses, and denoting by $\boldsymbol{x}_t^{\text{expert}}$ the prediction of one expert at time step $t$, we have

$$
\ell_t\big(\boldsymbol{x}_t^{\text{expert}}\big) = \big\langle\boldsymbol{g}_t,\boldsymbol{x}_t^{\text{expert}}\big\rangle = \big\langle g_t,\boldsymbol{x}_t^{\text{expert},(i_t)}\big\rangle \leq \|g_t\|_2\cdot\big\|\boldsymbol{x}_t^{\text{expert},(i_t)}\big\|_2 \leq DL_g.
$$

Hence, $L_\infty \leq DL_g$. Now, for any $\boldsymbol{u} \in \boldsymbol{V}$, the regret of Hedge-MT-OGD against $\boldsymbol{u}$ is upper bounded by the sum of: (1) the regret of Hedge with respect to the best expert in $\mathcal{C}_\varepsilon$, that is upper bounded by Equation (33), and (2) the regret of the best expert in $\mathcal{C}_\varepsilon$ against $\boldsymbol{u}$, that is upper bounded by Equation (32). Summing the two upper bounds and setting $\varepsilon = 1/N$ yields the desired result. □

## A.10 Proof of Proposition 10

Let $(\ell_{t,i})_{i\in\mathcal{A}_t}$ be the losses associated to the active agents at times step $t$. For MT-OMD run with matrix $A = A(b) = (1+b)I_N - \frac{b}{N}\mathbb{1}\mathbb{1}^\top$, for $b \geq 0$, and constant learning rate $\eta > 0$, we have

$$
\begin{aligned}
R_T(\boldsymbol{u}) &= \sum_{t=1}^T\sum_{i\in\mathcal{A}_t}\ell_{t,i}\big(\boldsymbol{x}_t^{(i)}\big) - \ell_{t,i}\big(\boldsymbol{u}^{(i)}\big) \\
&\leq \frac{\boldsymbol{u}^\top\boldsymbol{A}\boldsymbol{u}}{2\eta} + \frac{\eta}{2}\sum_{t=1}^T\|\boldsymbol{g}_t\|_{\boldsymbol{A}^{-1}}^2 \\
&= \frac{ND^2\big(1+b\frac{N-1}{N}\sigma^2\big)}{2\eta} + \frac{\eta}{2}\sum_{t=1}^T\|\boldsymbol{g}_t\|_{\boldsymbol{A}^{-1}}^2,
\end{aligned}
\tag{34}
$$

where $\boldsymbol{g}_t \in \mathbb{R}^{Nd}$ is the compound gradient vector with non-zero blocks at the indices present in $\mathcal{A}_t$. Recall that $A(b)^{-1} = \frac{1}{1+b}I_N + \frac{b}{1+b}\frac{\mathbf{1}\mathbf{1}^\top}{N}$. We have

$$\begin{aligned}
\|\boldsymbol{g}_t\|_{\boldsymbol{A}^{-1}}^2 &= \sum_{i,j} A_{ij}^{-1}\langle \boldsymbol{g}_t^{(i)}, \boldsymbol{g}_t^{(j)}\rangle \\
&= \sum_{i\in\mathcal{A}_t} A_{ii}^{-1}\|\boldsymbol{g}_t^{(i)}\|_2^2 + \sum_{i\neq j\in\mathcal{A}_t} A_{ij}^{-1}\langle \boldsymbol{g}_t^{(i)}, \boldsymbol{g}_t^{(j)}\rangle \\
&\leq \left(p\frac{b+N}{(1+b)N} + p(p-1)\frac{b}{(1+b)N}\right)L_g^2 \\
&= pL_g^2\frac{pb+N}{(1+b)N}
\end{aligned}$$

Substituting into equation 34 and setting $b = \sqrt{p}N$, we have

$$R_T(\boldsymbol{u}) \leq \frac{ND^2\left(1+\sqrt{p}\sigma^2(N-1)\right)}{2\eta} + \frac{\eta}{2}TL_g^2\frac{p(1+p)}{N}$$

Setting $\eta = \frac{ND}{L_g\sqrt{T}}\sqrt{\frac{1+\sqrt{p}\sigma^2(N-1)}{p(1+p)}}$, we finally get

$$R_t(\boldsymbol{u}) \leq DL_g\sqrt{pT}\sqrt{1+\sigma^2(N-1)}\sqrt{p^{1/2}+p^{3/2}}.$$

## B  Derivation of the Algorithms

In this appendix we gather all the technical details about the algorithms.

### B.1  Details on MT-OGD

With the change of feasible set, $\boldsymbol{x}_{t+1}$ produced by MT-OGD is the solution to

$$\begin{aligned}
\min_{\boldsymbol{x}\in\mathbb{R}^{Nd}} \quad & \|\boldsymbol{x}_t - \eta_t\boldsymbol{A}^{-1}\boldsymbol{g}_t - \boldsymbol{x}\|_{\boldsymbol{A}}^2, \\
\text{s.t.} \quad & \|\boldsymbol{x}\|_{\boldsymbol{A}}^2 \leq (1+b\sigma^2)ND^2,
\end{aligned}$$

or again

$$\begin{aligned}
\min_{\boldsymbol{x}\in\mathbb{R}^{Nd}} \quad & \|\boldsymbol{A}^{1/2}\boldsymbol{x}_t - \eta_t\boldsymbol{A}^{-1/2}\boldsymbol{g}_t - \boldsymbol{A}^{1/2}\boldsymbol{x}\|_2^2, \\
\text{s.t.} \quad & \|\boldsymbol{A}^{1/2}\boldsymbol{x}\|_2^2 \leq (1+b\sigma^2)ND^2.
\end{aligned}$$

Introducing the notation $\boldsymbol{y}_t = \boldsymbol{A}^{1/2}\boldsymbol{x}_t$, the update on the $\boldsymbol{y}_t$'s writes

$$\boldsymbol{y}_{t+1} = \mathrm{Proj}\left(\boldsymbol{y}_t - \eta_t\boldsymbol{A}^{-1/2}\boldsymbol{g}_t, \sqrt{(1+b\sigma^2)N}D\right).$$

### B.2  Details on MT-EG

Recall that for the negative entropy regularizer we have $B_\psi(\boldsymbol{x}, \boldsymbol{y}) = \sum_{i=1}^N\sum_{j=1}^d \boldsymbol{x}_j^{(i)} \ln\frac{\boldsymbol{x}_j^{(i)}}{\boldsymbol{y}_j^{(i)}}$. Expliciting the objective function of the first update, we obtain

$$\langle \eta\boldsymbol{A}^{-1/2}\boldsymbol{g}_t, \boldsymbol{y}\rangle + B_\psi(\boldsymbol{y}, \boldsymbol{y}_t) = \sum_{i=1}^N\sum_{j=1}^d \eta A_{ii_t}^{-1/2}g_{t,j}\,\boldsymbol{y}_j^{(i)} + \boldsymbol{y}_j^{(i)}\ln\frac{\boldsymbol{y}_j^{(i)}}{\boldsymbol{y}_{t,j}^{(i)}}.$$

For all $i \leq N$ and $j \leq d$, differentiating with respect to $\boldsymbol{y}_j^{(i)}$ and setting the gradient to 0 we get

$$\tilde{\boldsymbol{y}}_{t+1,j}^{(i)} = \boldsymbol{y}_{t,j}^{(i)} \, \exp\left(-\eta A_{ii_t}^{-1/2} g_{t,j} - 1\right). \tag{35}$$

We focus now on the second updateEquation (17). The constraint $\boldsymbol{y} \in \boldsymbol{A}^{1/2}(\boldsymbol{\Delta})$ rewrites $\boldsymbol{A}^{-1/2}\boldsymbol{y} \in \boldsymbol{\Delta}$, or equivalently

$$\forall i \leq N, \qquad \mathbb{1}^\top \left(\boldsymbol{A}^{-1/2}\boldsymbol{y}\right)^{(i)} = 1,$$

$$\forall i \leq N, \; j \leq d, \qquad \left(\boldsymbol{A}^{-1/2}\boldsymbol{y}\right)_j^{(i)} \geq 0.$$

We introduce matrix $Y \in \mathbb{R}^{N \times d}$ such that $Y_{kl} = \boldsymbol{y}_l^{(k)}$. Then it holds for all $i \leq N$

$$\mathbb{1}^\top \left(\boldsymbol{A}^{-1/2}\boldsymbol{y}\right)^{(i)} = \sum_{l=1}^d \left(\boldsymbol{A}^{-1/2}\boldsymbol{y}\right)_l^{(i)} = \sum_{l=1}^d \sum_{k=1}^N A_{ik}^{-1/2} Y_{kl} = \sum_{l=1}^d \left[A^{-1/2}Y\right]_{il} = \left[A^{-1/2}Y\mathbb{1}\right]_i.$$

Hence, using that $A^{-1/2}$ is stochastic, the first constraint rewrites

$$A^{-1/2}Y\mathbb{1} = \mathbb{1} \iff Y\mathbb{1} = \mathbb{1} \iff \forall i \leq N, \quad \mathbb{1}^\top \boldsymbol{y}^{(i)} = 1. \tag{36}$$

Similarly, the second constraint reads:

$$\forall k \leq N, \; j \leq d, \qquad \left[A^{-1/2}Y\right]_{kj} \geq 0. \tag{37}$$

The Lagrangian associated to Problem Equation (17) writes

$$\mathcal{L}(\boldsymbol{y}, \Lambda, \boldsymbol{\mu}) = \sum_{i=1}^N \sum_{j=1}^d \boldsymbol{y}_j^{(i)} \ln \frac{\boldsymbol{y}_j^{(i)}}{\tilde{\boldsymbol{y}}_{t+1,j}^{(i)}} - \sum_{k=1}^N \sum_{j=1}^d \Lambda_{kj}\left[A^{-1/2}Y\right]_{kj} + \sum_{i=1}^N \mu_i(\mathbb{1}^\top \boldsymbol{y}^{(i)} - 1)$$

$$= \sum_{i=1}^N \sum_{j=1}^d Y_{ij} \ln \frac{Y_{ij}}{\tilde{\boldsymbol{y}}_{t+1,j}^{(i)}} - \sum_{k=1}^N \sum_{j=1}^d \sum_{i=1}^N \Lambda_{kj} A_{ki}^{-1/2} Y_{ij} + \sum_{i=1}^N \sum_{j=1}^d \mu_i Y_{ij} - \mathbb{1}^\top \boldsymbol{\mu},$$

with $\Lambda \in \mathbb{R}^{N \times d}$ and $\boldsymbol{\mu} \in \mathbb{R}^N$ the Lagrange multipliers associated to constraints Equation (37) and Equation (36) respectively. For all $i \leq N$, $j \leq d$, differentiating with respect to $Y_{ij}$ and setting the gradient to 0 yields

$$\left[Y^{(t+1)}\right]_{ij} := \boldsymbol{y}_{t+1,j}^{(i)} = \tilde{\boldsymbol{y}}_{t+1,j}^{(i)} \, e^{\left[A^{-1/2}\Lambda\right]_{ij} - \mu_i - 1}. \tag{38}$$

Furthermore, the complementary slackness writes for all $i \leq N$ and $j \leq d$

$$\Lambda_{ij}\left[A^{-1/2}Y^{(t+1)}\right]_{ij} = 0.$$

However, Equation (38) gives $\left[Y^{(t+1)}\right]_{ij} > 0$, and matrix $A^{-1/2}$ is stochastic (see Lemma 13), so we have $\left[A^{-1/2}Y^{(t+1)}\right]_{ij} > 0$, and consequently $\Lambda_{ij} = 0$. Then

$$\boldsymbol{y}_{t+1,j}^{(i)} = \tilde{\boldsymbol{y}}_{t+1,j}^{(i)} \, e^{-\mu_i - 1}.$$

Using $\mathbb{1}^\top \boldsymbol{y}_{t+1}^{(i)} = 1$, we get $e^{-\mu_i - 1} = 1/\left(\sum_{j=1}^d \tilde{\boldsymbol{y}}_{t+1,j}^{(i)}\right)$. Substituting Equation (35) we get for all $i \leq N$

$$\boldsymbol{y}_{t+1,j}^{(i)} = \frac{\tilde{\boldsymbol{y}}_{t+1,j}^{(i)}}{\sum_{k=1}^d \tilde{\boldsymbol{y}}_{t+1,k}^{(i)}}$$

$$= \frac{\boldsymbol{y}_{t,j}^{(i)} \, e^{-\eta A_{ii_t}^{-1/2} g_{t,j} - 1}}{\sum_{k=1}^d \boldsymbol{y}_{t,k}^{(i)} \, e^{-\eta A_{ii_t}^{-1/2} g_{t,k} - 1}}$$

$$= \frac{\boldsymbol{y}_{t,j}^{(i)} \, e^{-\eta A_{ii_t}^{-1/2} g_{t,j}}}{\sum_{k=1}^d \boldsymbol{y}_{t,k}^{(i)} \, e^{-\eta A_{ii_t}^{-1/2} g_{t,k}}}.$$

**Lemma 13.** *Let $A = I_N + L$, where $L$ is the Laplacian matrix associated to any weighted undirected graph on $\{1, \ldots, N\}$. Then $A^{-1}$ and $A^{-1/2}$ are (doubly) stochastic matrices.*

*Proof.* It is immediate to see from the definition of $A$ that $A^{-1}$ and $A^{-1/2}$ are both symmetric and satisfy $A^{-1}\mathbb{1} = A^{-1/2}\mathbb{1} = \mathbb{1}$. It remains to check that their entries are nonnegative. To that end, we use that inverses of $M$-matrices are entrywise nonnegative. Matrix $A$ is a non-singular $M$-matrix, as its off-diagonal entries are nonpositive (i.e., $A$ is a $Z$-matrix) and its eigenvalues are positive. As a consequence, $A^{1/2}$ is also a $M$-matrix (Alefeld & Schneider, 1982, Theorem 4), which concludes the proof. $\qquad\square$

## C   Technical Comparison to Cavallanti et al. (2010)

First, we would like to remind the reader that, unlike Cavallanti et al. (2010): ($i$) we tackle any subdifferentiable loss function, and not only the hinge loss for classification, ($ii$) we address any strongly convex regularizer, and not only the squared Euclidean norm, ($iii$) our proofs provide clear insights on the regularizer's behaviour, and are not just black box applications of the Kernel Perceptron Theorem, ($iv$) we provide (matching) lower bounds, ($v$) we develop what we believe is the correct generalization to $p$-norms, see technical details below, ($vi$) we provide a unifying framework and a general analysis that allow to deal with non-Euclidean geometries on the simplex, ($vii$) we are adaptive to the task variance $\sigma^2$.

About the $p$-norm extension, the two regularizers used in Cavallanti et al. (2010) and this paper are fundamentally different, as they write respectively

$$\boldsymbol{\psi}(\boldsymbol{u}) = \|\boldsymbol{Au}\|_p^2, \qquad \text{and} \qquad \boldsymbol{\psi}(\boldsymbol{u}) = \sum_{i=1}^{N} \left\| (\boldsymbol{A}^{1/2}u)^{(i)} \right\|_p^2 .$$

A first advantage of our method is that we recover the Euclidean framework by setting $p = 2$, which is not the case for the regularizer used in Cavallanti et al. (2010). This makes us believe that we propose the right way to address $p$-norms, and that a general theory of multitask acceleration is worth developing to avoid misinterpretations of this kind. A second advantage is that our bounds have a much better dependence with respect to the task variance $\sigma^2$ and the number of tasks $N$. Recall that after several approximations, see Appendix A.5, our bound for $p$-norms on the simplex features the variance term

$$\sqrt{1 + (N-1)\sigma^2} \leq 1 + \sqrt{N}\sigma , \tag{39}$$

where $\sigma^2$ is an upper bound of the task variance according to $\|\cdot\|_1$, such that

$$\text{Var}_{\|\cdot\|_1}(\boldsymbol{u}) = \frac{1}{N-1} \sum_{i=1}^{N} \left\| \boldsymbol{u}^{(i)} - \bar{\boldsymbol{u}} \right\|_1^2 \leq \sigma^2 .$$

On the other hand, the bound of Theorem 6 in Cavallanti et al. (2010) features the variance term

$$\frac{\|\boldsymbol{Au}\|_1}{N} = \frac{1}{N} \sum_{i=1}^{N} \left\| (1+N)\boldsymbol{u}^{(i)} - N\bar{\boldsymbol{u}} \right\|_1$$

$$\leq \frac{1}{N} \sum_{i=1}^{N} \left( \left\| \boldsymbol{u}^{(i)} \right\|_1 + N \left\| \boldsymbol{u}^{(i)} - \bar{\boldsymbol{u}} \right\|_1 \right)$$

$$\leq 1 + \sum_{i=1}^{N} \left\| \boldsymbol{u}^{(i)} - \bar{\boldsymbol{u}} \right\|_1$$

$$\leq 1 + \sqrt{N \sum_{i=1}^{N} \left\| \boldsymbol{u}^{(i)} - \bar{\boldsymbol{u}} \right\|_1^2}$$

$$\leq 1 + N\sigma . \tag{40}$$

From Equations (39) and (40), it is clear that our bounds exhibit a much better dependence with respect to $\sigma^2$ and N. We further highlight that the bound of Theorem 6 in Cavallanti et al. (2010) contains an extra term which is the square of Equation (40), without being multiplied by any time-related quantity though.

## D  Additional Experiment

We report the results of a synthetic experiment where we plot the multitask regret of MT-OGD against the standard deviation $\sigma$ of the reference vectors. Specifically, we consider a regression problem in $\mathbb{R}^2$ with respect to the square loss. We set $D = 1$ and generate $N = 4$ tasks defining the reference vectors as in the proof of Proposition 3 (see Appendix A.3), so that the task variance is set to the desired value. Each task consists of a sequence of 10 losses, so that $T = 40$. The losses for the $i$-th task are generated by first sampling an instance $x$ from the unit sphere centered at $\mathbf{1}$, and then computing its label as $y = \langle \boldsymbol{u}^{(i)}, x \rangle + \epsilon$, where $\epsilon$ is an independent Gaussian noise with variance 0.1 and truncated in [-1, 1]. We plot the final multitask regret $R_T$ averaged over 30 runs. MT-OGD is tuned with the optimal parameter choices for $b$ and $\eta$ according to the theory. Results are shown in Figure 2. As expected, the regret of MT-OGD increases linearly with $\sigma$, as suggested by the upper bound Equation (14) and the lower bound in Proposition 3. For the sake of comparison, we also benchmark IT-OGD and report its best average performance. The switch in performance occurs slightly before $\sigma = 1$, which is also expected as, aside from the dependence in $\sigma$, the bounds for MT-OGD scales with $\sqrt{2T}$ (instead of $\sqrt{T}$ for IT-OGD) and are thus slightly worse for $\sigma = 1$.

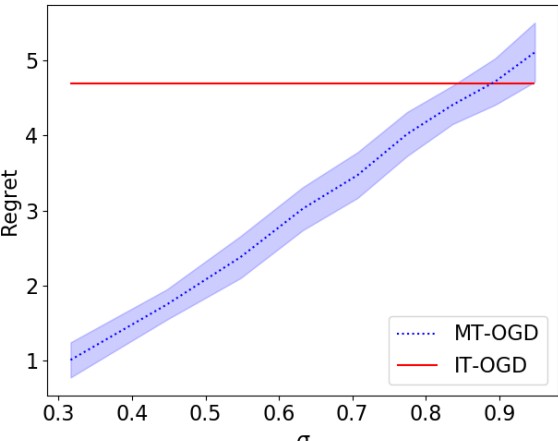

Figure 2: Multitask regret $R_T$ of MT-OGD and IT-OGD for $T = 40$ against the standard deviation $\sigma$.

