# OpenReview forum: "Multitask Online Mirror Descent"
_TMLR — Accepted by TMLR_

### Review · Reviewer_mLuu · 2022-06-29

**Summary Of Contributions:**

This work proposes a multitask generalization of the online mirror descent method, which is able to exploit the correlation between tasks, by sharing updates between tasks. Specifically, the authors derive the MT-OGD method by introducing a interaction matrix in the standard OGD method and show that with a particular choice of the interaction, MT-OGD enjoys a multitask acceleration under the assumption that the variance of the reference vectors is small. While the above result is derived for the Euclidean regularization, the authors show that similar results can be obtained in general norm regularizer and Bregman divergence settings. Finally, a Hedge-based extension of MT-OMD is proposed which adaptive to the task variance parameter with the regret increased by only a small additional factor. The authors show that MT-OGD and MT-EG can be carried out efficiently and conduct simple experiments to validate their theoretical results empirically.

**Broader Impact Concerns:**

This work may be a good starting point to provably show that the collaboration in FL is actually beneficial to users, which, in the first place, is the incentive of sharing private information.

**Requested Changes:**

I do not have much to complain about this paper and believe it can be readily accepted in the current status.

**Strengths And Weaknesses:**

Strength:
This paper is very well written. The introduction of the multitask extension of the OGD method is intuitive and the different interpretations (8)-(10) of the MT-OGD update (7) make it easier for the reader to understand the derivation.
I find the result of this paper novel and timely: In federate learning, it is crucial to provide the justification of the necessity of collaboration since sharing information with others inevitably causes privacy issue and should only be conducted when there are actual benefits. I believe this work provides a solid stepping stone for analyzing the federated learning case.
The obtained theoretical improvement, coined multitask acceleration, is intuitive and the authors provide a complete picture on how such an improvement can be extended to general norm regularizer and Bregman divergence cases.
Finally, the author also provided a variant of MT-OMD which is adaptive to the task variance parameter which is critical from a practical consideration.

Weakness:
I do not have much to complain about this paper and believe it can be readily accepted in the current status.

---

> ### Author Response · Authors · 2022-07-20
> **Response to Reviewer mLuu**
>
> We thank the reviewer for her/his very positive feedback!

---

### Review · Reviewer_vbu3 · 2022-07-08

**Summary Of Contributions:**

1. This paper develops a general framework for multitask online convex optimization;
2. The authors propose a variant of OMD that can adapt to the variance of the compartors of different tasks;
3. The authors also consider several extensions, including a matching lower bound, measuring the variances with different norms, and how to deal with the case where the parameter of variance is agnostic;
4. Experimental results on real-world data demonstrate the effectiveness of the proposed method.


**Broader Impact Concerns:**

This is a theoretical paper and does not have any ethical concerns.

**Requested Changes:**

1. It seems to me that the regret defined in this paper (eq. 1) is related to dynamic regret and adaptive regret in online convex optimization. Indeed, for example, minimizing dynamic regret can be considered as doing T tasks, and the algorithm for dynamic regret can adapt to the change of comparators (V_T=\sum_{t=1}^T ||u_{t+1}-u_{t}||). It would be great if the authors could add more discussion to this. On the other hand, I was wondering if the proposed algorithm can also be used for minimizing dynamic regret and obtain a simga-dependant bound?

2. To make the paper more easier to read, it would be better if the authors could write down the pudo-code for the algorithm more explicitly, using \begin{algorithm} \end{algorithm}.

**Strengths And Weaknesses:**

Strengths:
1. This paper formulates a general paradigm for mullti-task OCO, which is an important and well-motivated problem;
2. The proposed algorithm successfully achieves a tighter regret bound wrt the number of tasks when the tasks are similar (in the sense that the variance sigma of the compators are is small). To my knowledge, the idea of adapting to the variance of u is novel and interesting.
3. The authors did a thorough work for the proposed problem, and studied various extensions, which are mentioned in the summary above.

Weakness:
1. The vanilla algorithm requires to know sigma as a prior knowledge. Although the authors propose a hedge version of the algorithm which is agnostic to sigma, there is an extra root(logN T) factor, which exists even if sigma is 0.
2. Currently the configuration of A is fixed before the learning process, which means that it cannot adapt to the data like Adagrad or Adam.
3. It would be great if the authors could extend the proposed algorithm to adapt to other types of loss functions such as exp-concave and strongly convex functions.

---

> ### Author Response · Authors · 2022-07-20
> **Response to Reviewer vbu3**
>
> We thank the reviewer for her/his feedback. Please find below our answers to the weaknesses (W1 - W3) and requested changes (RC 1, RC 2) pointed out.
>
> **W1.** The knowledge of $\sigma$ acts as a learning bias and/or prior knowledge on the problem. Obviously, as there is no free lunch, if the learner has no clue about the problem to solve, it has to resort to a richer model class (in this case the set of algorithm’s instances, indexed by $\sigma$) which in turn results in a larger regret. Note that even if $\sigma = 0$, one cannot hope for paying less than a $\sqrt{T}$ regret (lower bound for single task), such that the additive $\sqrt{T \log N}$ is essentially of the same order, and seems a reasonable overhead in exchange of being adaptive to $\sigma$.
>
> **W2.** Similarly to the choice of sigma, the knowledge of the similarity graph can be seen as prior knowledge. Learning the similarity matrix is however a significantly harder problem: we have to learn $N^2$ interaction coefficients instead of a single scalar value in $[0, 1]$. As a consequence, a simple covering approach as the one used to learn $\sigma$ is not possible, since the space of candidate interaction matrices is too large. We further highlight that the optimal interaction matrix here depends on the **unknown** optimal task vectors that we are currently learning, and not on external information (such as the gradients magnitudes in AdaGrad), which makes the analysis significantly more intricate. We leave this interesting question as future research.
>
>
> **W3.** Extending our analysis to strongly convex functions is unfortunately not possible. Indeed, the cornerstone of our analysis is to leverage the compound representation in which the interactions are more easily analyzed. On the other hand, the strong convexity of the losses provides a sharper upper bound on the instantaneous regret that depends on the norm of the active predictor/comparator only, but cannot be extended to the compound framework. Formally, we cannot write (although it would be necessary for an analysis)
>
> $$
> \ell\_t(x_t) - \ell\_t(u) \le \langle g\_t, x\_t - u\rangle - \frac{\mu}{2} \\|x_t - u\\|\_2^2 ~~\cancel{\le}~ \langle \mathbf{g}\_t, \mathbf{x}\_t - \mathbf{u}\rangle - \frac{\mu}{2} \\|\mathbf{x}\_t - \mathbf{u}\\|\_{\mathbf{A}}^2
> $$
>
> Another way to look at the problem is to recall that FTRL deals with strongly convex losses by choosing $\psi \equiv 0$. This means for us $\tilde{\mathbf{\psi}} = \mathbf{\psi}\big(\mathbf{A}^{1/2} \cdot\big)\equiv 0$, such that MT-FTRL is equivalent to independent FTRL and no multitask improvement can be achieved.
>
> Exp-concave losses suffer from the same problem. Let $\ell\_t$ be $\alpha$ exp-concave, we do not have
>
> $$
> \ell\_t(x_t) - \ell\_t(u) \le \langle g_t, x_t - u\rangle - \frac{\alpha}{4} \langle g_t, x_t - u\rangle^2 ~~\cancel{\le}~ \langle \mathbf{g}\_t, \mathbf{x}\_t - \mathbf{u}\rangle  - \frac{\alpha}{4} \langle \mathbf{g}\_t, \mathbf{x}\_t - \mathbf{u}\rangle\_{\mathbf{A}}^2
> $$
>
> Note that even if the above inequality held, the parameter $b$ (whose optimization yields the multitask acceleration) would appear inside the log term, preventing from obtaining significant improvements. The entire discussion will be added.
>
> **RC 1.** Indeed, our setting can be viewed as a special case of dynamic regret, where each comparator $u_t$ in the sequence $u_1,u_2,\ldots$ belongs to an unknown set $\{u_1',\ldots,u_N'\}$ of known cardinality $N$. Moreover, at the beginning of each time step $t$, the learner is told the index $i_t \in \{1,\ldots,N\}$ of the comparator $u_t$ against which the regret is measured at time $t$. Therefore, our algorithms cannot be used to solve a generic dynamic regret minimization problem. On the other hand, any algorithm for dynamic regret minimization can be used in our setting. In the Euclidean case, the optimal dynamic regret bound is $\sqrt{D^2 + D V_T}\sqrt{T}$, where $D$ is the Euclidean diameter of the decision space. In order to facilitate a comparison with our bound $\sqrt{D^2 + D^2(N-1)\sigma^2}\sqrt{T}$, let assume that the sequence of adversarial activations is such that $i_t \neq i_{t-1}$ and that all pairs of distinct elements in $\{u_1',\ldots,u_N'\}$ appear with the same frequency as consecutive comparators $u_t,u_{t+1}$. Let $P_T = \sum_{t=1}^T ||u_{t+1}-u_t||_2^2$. We have
>
> $$
> P_T = \sum\_{t=1}^T ||u_{t+1}-u_t||_2^2 = \frac{T}{N(N-1)} \sum\_{i \ne j} ||u'_i - u'_j||_2^2 = 2T \frac{1}{N(N-1)} \sum\_{i \ne j} \frac{||u'_i - u'_j||_2^2}{2} = 2T \sigma^2,
> $$
>
> such that our upper bound is at most
> $$
> \sqrt{D^2 + D^2(N-1)\sigma^2}\sqrt{T} = \sqrt{D^2 + D^2(N-1)\frac{P_T}{2T}}\sqrt{T} \le \sqrt{D^2 + \frac{D^2(N-1)}{2T} DV_T}\sqrt{T},
> $$
> which is better as soon as $T \ge \frac{D^2(N-1)}{2}$. We will add a discussion about this point.
>
> **RC 2.** Thank you for the suggestion, we will add the pseudo-code of the generic algorithm and of the two instances (MT-OGD and MT-EG).

---

### Review · Reviewer_vFPH · 2022-07-10

**Summary Of Contributions:**


This paper initiates the study of multitask online learning problem. Here, $N$ agents are online-learning $N$ different tasks. The agents and the environment interact for $T$ time steps in total. At each time step $t \in [T]$, the environment can pick an agent $i_t$ and require it to make a prediction $x_t$, the environment then chooses a convex loss function $\ell_t$. The agent $i_t$ incurs a loss $\ell_t(x_t)$ and observes a subgradient $\nabla \ell_t(x_t)$. The goal is to minimize the multitask regret, defined as the sum of individual regret. Naively running $N$ independent OMD would result in $O(\sqrt{NT})$ regret. In this paper, the authors show that by communication across agents, the regret could be reduced to $O(\sqrt{1 + \sigma^2(N-1)} \sqrt{T})$, where $\sigma^2$ is the sample variance of the ground truth of all agents.

Furthermore, this paper proves a matching lower bound, showing that the results are near-optimal. This paper begins with an algorithm that only works with known $\sigma^2$ and $\ell_2$ regularizer, while they further generalize to $\ell_p$ norm regularizers. This paper also provides experiments on synthetic and real-world datasets to cooroborate their theoretical results.


**Broader Impact Concerns:**

This paper is mainly theoretical and I don't see any potential negative ethical impact.

**Requested Changes:**


1. Prop. 3: could you state $D$ and $L_g$ more precisely? I'm assuming the values are the same as in Prop. 2.
2. Setting: could you make it more clear on whether the environment selects $\ell_t$ after seeing $x_t$?


**Strengths And Weaknesses:**


### Strengths

1. This paper comes up with a novel setting of multitask online learning.
1. With a matching upper and lower bound, this paper almost settles their multitask online learning problem.
2. Experimental results matches theoretical results. Importantly, the authors verified the dependency on the $\sigma^2$ parameter.
3. The paper is generally well-organized and the writing is good. I did not find typo.
4. The paper provides a summary of notations and symbols in the Appendix, which facilitates readers.

### Weakness

1. Mirror descent is a framework of algorithms. Its main feature is that it covers analyses for different regularizers, in particular the KL divergence regularizer. However, the authors seem not to mention if the algorithm could work for KL regularizers.
2. The formulation only requires one agent to select $x_t$, and is allowed to select agent adversarially. I personally think this may require more motivation, because in multitask bandit learning literature, I usually see settings where all agents are required to play.

---

> ### Author Response · Authors · 2022-07-20
> **Response to Reviewer vFPH**
>
> We thank the reviewer for her/his feedback. Please find below our answers to the weaknesses (W1, W2) and requested changes (RC 1, RC 2) pointed out.
>
> **W1.** Indeed, Online Mirror Descent covers several algorithms, and we showed that multitask acceleration can be achieved with different regularizers, such as the square Euclidean norm (Proposition 2) or general square norms (Proposition 5). Section 3.3 in then devoted to regularizers defined on the simplex. In particular, Proposition 8 holds for any regularizer $\psi$ defined on the simplex $\Delta$ for which there exist $x^* \in \Delta$ and $C>0$ such that $B_\psi(x, x^*) \le C$ for any $x \in \Delta$. As specified between the two displayed equations of Proposition 8, the negative entropy (which induces the KL divergence as Bregman divergence) satisfies this assumption with $x^* = \mathbf{1}/d$ and $C=\ln d$. The last displayed equation then states the regret guarantee for our multitask extension of the KL regularizer.
>
> **W2.** Adversarial activations are more flexible as they account for scenarios in which one task (e.g., one user in a federated learning framework) is active more often that the others. The case where all agents are active can be reduced to cyclic adversarial activations. Indeed, consider the multitask extension of FTRL (with predictions every multiple of $N$ activations) after $t = k N$ time steps composed of $k$ rounds of cyclic activations (with $\bar{g}_{\tau, i}$ being the subgradient for task $i$ during round $\tau \le k$). We have
>
> \begin{align*}
> \mathbf{x}_{t+1} = \text{argmin}\_\{\mathbf{x} \in \mathbf{V}\} ~ \mathbf{\psi}\big(\mathbf{A}^{1/2}\mathbf{x}\big) + \eta \bigg\langle \sum\_{s=1}^{t} \bar{g}\_s, \mathbf{x}\bigg\rangle = \text{argmin}\_\{\mathbf{x} \in \mathbf{V}\} ~ \mathbf{\psi}\big(\mathbf{A}^{1/2}\mathbf{x}\big) + \eta \bigg\langle\sum\_{\tau=1}^k \left(\sum\_{i=1}^N \bar{g}\_{\tau, i}\right), \mathbf{x}\bigg\rangle
> \end{align*}
>
> Here, only the sum of the gradients seen so far matters. Whether they are collected one after the other in a cyclic fashion like in the first case, or by group of simultaneous activations like in the second one, does not change the prediction. Actually, the only requirement for the task activations is that at each $t = kN$, all tasks have been activated exactly $k$ times, independently from their order among two different rounds. Note that OMD does not enjoy the same equivalence due to the intermediate projections. We will add this discussion.
>
> **RC 1.** Indeed, $D$ is the diameter of the set of individual predictors and $L_g$ is the Lipschitz constant of the losses. We will clarify this in the revision.
>
> **RC 2.** Since our algorithm is deterministic, this makes no difference in our case. This will be stated explicitly.

---

> > ### Comment · Reviewer_vFPH · 2022-07-28
> > **Reply**
> >
> > Thank the authors for the thorough feedback!

---

### Decision · Action_Editors · 2022-08-08

**Recommendation:** Accept as is

**Comment:**

The paper got overall positive reviews and all reviewers voted for acceptance.
I think this paper is well within the scope of TMLR and should be accepted.